# Information-Theoretic Generalization Bounds for SGLD via Data-Dependent Estimates

**Jeffrey Negrea**[*]
University of Toronto,
Vector Institute

**Mahdi Haghifam**[*]
University of Toronto,
Element AI

**Gintare Karolina Dziugaite**
Element AI

**Ashish Khisti**
University of Toronto

**Daniel M. Roy**
University of Toronto,
Vector Institute

## Abstract

In this work, we improve upon the stepwise analysis of noisy iterative learning algorithms initiated by Pensia, Jog, and Loh (2018) and recently extended by Bu, Zou, and Veeravalli (2019). Our main contributions are significantly improved mutual information bounds for Stochastic Gradient Langevin Dynamics via data-dependent estimates. Our approach is based on the variational characterization of mutual information and the use of data-dependent priors that forecast the mini-batch gradient based on a subset of the training samples. Our approach is broadly applicable within the information-theoretic framework of Russo and Zou (2015) and Xu and Raginsky (2017). Our bound can be tied to a measure of flatness of the empirical risk surface. As compared with other bounds that depend on the squared norms of gradients, empirical investigations show that the terms in our bounds are orders of magnitude smaller.

## 1   Introduction

Stochastic subgradient methods, especially stochastic gradient descent (SGD), are at the core of recent advances in deep-learning practice. Despite some progress, developing a precise understanding of generalization error for that class of algorithms remains wide open. Concurrently, there has been steady progress for noisy variants of SGD, such as stochastic gradient Langevin dynamics (SGLD) [13, 26, 34] and its full-batch counterpart, the Langevin algorithm [13]. The introduction of Gaussian noise to the iterates of SGD expands the set of theoretical frameworks that can be brought to bear on the study of generalization. In pioneering work, Raginsky, Rakhlin, and Telgarsky [26] exploit the fact that SGLD approximates Langevin diffusion, a continuous time Markov process, in the small step size limit. One drawback of this and related analyses involving Markov processes is the reliance on mixing. We hypothesize that SGLD is not mixing in practice, so results based upon mixing may not be representative of empirical performance.

In recent work, Pensia, Jog, and Loh [24] perform a stepwise analysis of a family of noisy iterative algorithms that includes SGLD and the Langevin algorithm. At the foundation of this work is the framework of Russo and Zou [29] and Xu and Raginsky [35], where mean generalization error is controlled in terms of the mutual information between the dataset and the learned parameters. (See also the study of on-average KL stability by Wang, Lei, and Fienberg [33].) However, because the data distribution is unknown, so is any mutual information involving the data. This presents a significant barrier to understanding generalization in terms of mutual information.

---

[*]Equal contribution authors, order of names was determined randomly.

One of the key contributions of Pensia et al. is a bound on the mutual information between the data and the final weights, which they construct from a bound on the mutual information between the data and the entire trajectory of weights. By exploiting properties of mutual information, they express the latter as a sum of conditional mutual informations associated with each gradient step. While these conditional mutual informations are also unknown, Pensia et al. obtain a bound in terms of the Lipschitz constant for the objective function being optimized.

By passing to the full trajectory and exploiting Lipschitz continuity, Pensia et al. circumvent the statistical barrier posed by the unknown mutual information. Their analysis, however, introduces several sources of looseness. In particular, the use of Lipschitz constants, which lead to distribution-*independent* bounds, eradicates any hope that these bounds will be non-vacuous for modern models and datasets. Indeed, for deep neural networks, the Lipschitz constant for the empirical risk would be prohibitively large, or in some cases infinite, and would immediately render any bound that depends on them vacuous in regimes of interest. In order to fully exploit the decomposition proposed by [24], one needs distribution-*dependent* bounds on the incremental mutual information at each step.

In fact, by a small change, the bounds established by Pensia et al. can be made to depend on expected-squared-gradient-norms, rather than Lipschitz constants, producing distribution-dependent bounds. The resulting bound would be similar to a PAC-Bayesian bound due to Mou et al. [22], which we consider to be the SGLD generalization result most similar to the present work. Writing $\sum_{t \leq T} \eta_t$ for $\sum_{t=1}^{T} \eta_t$, their bound is $O\big(\sqrt{(\beta/n)\sum_{t \leq T} \eta_t}\big)$ and does not place restrictions on the learning rate or Lipschitz continuity of the loss or its gradient. In other related work, Li, Luo, and Qiao [20] derive an $O\big((1/n)\sqrt{\beta \sum_{t \leq T} \eta_t}\big)$ generalization bound for SGLD that depends on expected-squared-gradient-norms. However their result requires the learning rate to scale inversely with the inverse temperature and the Lipschitz constant of the loss, severely limiting the applicability of their result to typical learning problems. Empirically, squared gradient norms are very large during training, which suggests that bounds based on these quantities may not explain empirical performance. As we will show, the dependence on the expected-squared-gradient-norm is spurious.

The key contribution of the present work is the observation that variants of the mutual information between the learned parameters and a subset of the data can be estimated using the rest of the data. We refer to such estimates as *data-dependent* due to their intermediate dependence on part of the data. The use of data-dependent estimates leads to distribution-dependent bounds that naturally adapt to the model of interest and the data distribution. In particular, using data-dependent estimates, we arrive at bounds in terms of the *incoherence* of gradients in the dataset. Roughly speaking, the incoherence measures the amount by which batch gradients computed on subsets of the data disagree, as quantified by squared norm. Crucially, the incoherence is never larger than the squared-gradient-norm on average, and the incoherence is 0 for most iterations of SGLD with small batches.

We note that the mutual information between learned parameter and a single data point is used to produce generalization bounds in work by Bu, Zou, and Veeravalli [6], Raginsky et al. [27], and Wang, Lei, and Fienberg [33]. However, in the SGLD analysis of [6], they do not use data-dependent estimates. Instead, they also rely on Lipschitz constants, leading to bounds similar to [24].

In the process of developing tighter distribution-dependent bounds, we also observe that, in some circumstances, one may obtain tighter estimates by working with conditional or disintegrated information-theoretic quantities. In particular, doing so provides more opportunities to exchange expectation and concave functions than are available with previous mutual information bounds. Using their own mutual information bound and the chain rule, [6] improve on the generalization error bound for SGLD from [24] by a factor of $\sqrt{\log n}$ where $n$ is the sample size. The advantage of [6] that enables this improvement is that their bound is only penalized once per epoch at a randomly chosen step. This effectively changes the order of an expectation and square-root, improving the bound. Building upon [6, 29, 35], we develop generalization bounds in terms of disintegrated information-theoretic quantities that extract expectations from concave functions as much as possible.

Finally, much like the stepwise analysis of SGD carried out by Hardt, Recht, and Singer [14], one could consider an analysis in terms of uniform stability, e.g., in terms of average leave-one-out KL stability [12]. Under an assumption of uniform stability, [22] also showed that expected generalization error decays rapidly at a $O(1/n)$ rate. However, uniform stability has poor dependence on the Lipschitz constant, and so, does not even hold in simple settings, like univariate logistic regression. As such, we do not believe this framework is suitable for studying SGLD as applied in modern

machine learning. For other work on information-theoretic analyses generalization error, and on SGLD, see [1, 3, 4, 15, 16, 27, 32].

## 1.1 Contributions

The present paper makes the following contributions:

- We provide novel information-theoretic generalization bounds that relate a learned parameter to a random subset of the training data. These bounds depend on forms of on-average information stability, but are different from those in existing work due to our use of disintegration.

- We introduce the technique of data-dependent priors for bounding mutual information in data-dependent estimates of expected generalization error. Specifically, we use data-dependent priors to forecast the dynamics of iterative algorithms using a randomly chosen subset of the data. Each possible subset yields a generalization bound for the empirical risk over the complementary subset. Combining this with our information-theoretic generalization bounds, we recover generalization error bounds for the empirical risk on the full dataset.

- We develop bounds for Langevin dynamics and SGLD that depend on a measure of the *incoherence* of empirical gradients. This quantity is typically orders of magnitude smaller than the squared gradient norms or Lipschitz constants that other bounds depend upon. In our experiments, the difference was a multiplicative factor between $10^2$ and $10^4$.

- Our generalization bound for SGLD is $O\big(\min\big\{\sqrt{(\beta/bn)\sum_{t\leq T}\eta_t},\ (1/n)\sum_{t\leq T}\sqrt{\beta\eta_t}\big\}\big)$ where $\eta_t$ is the learning rate at iteration $t$, $T$ is the number of iterations, $\beta$ is the inverse temperature, and $b$ is the minibatch size. This bound is currently state of the art for bounds without assumptions on the smoothness of the loss or restrictions on the learning rate.

## 1.2 Preliminaries

Let $\mathscr{D}$ be an unknown distribution on a space $\mathscr{Z}$ and let $\mathscr{W}$ be a space of parameters. Consider a loss function $\ell: \mathscr{Z} \times \mathscr{W} \to \mathbb{R}$ and the corresponding risk function $R_{\mathscr{D}}(w) = \mathbb{E}\ell(Z,w)$. Given an i.i.d. dataset of size $n$, $S \sim \mathscr{D}^n$, we may form the empirical risk function $\hat{R}_S(w) = \frac{1}{n}\sum_{i=1}^m \ell(Z_i,w)$, where $S = (Z_1, \ldots, Z_n)$. In the setting of classification and continuous parameter spaces, the loss function is discontinuous and the empirical risk function does not convey useful gradient information. For this reason, it is common to work with a *surrogate* loss, such as cross entropy. To that end, let $\tilde{\ell}: \mathscr{Z} \times \mathscr{W} \to \mathbb{R}$ denote a surrogate loss and let $\tilde{R}_{\mathscr{D}}(w) = \mathbb{E}\tilde{\ell}(Z,w)$ and $\tilde{R}_S(w) = \frac{1}{n}\sum_{i=1}^m \tilde{\ell}(Z_i,w)$ be the corresponding surrogate risk and empirical surrogate risk.

Our primary interest is in the generalization performance of learning algorithms. Abstractly, let $W$ be a random element in $\mathscr{W}$ satisfying $W = \mathscr{A}(S,V)$, where $V$ is some auxiliary random element independent from $S$ and $\mathscr{A}$ is a measurable function representing a randomized learning algorithm that maps the data $S$ to a learned parameter $W$. Our focus will be the **(mean) generalization error** of $W$, i.e., $\mathbb{E}\big[R_{\mathscr{D}}(W) - \hat{R}_S(W)\big]$. Note that we have averaged over both the choice of dataset and the source of randomness $V$ available to the learning algorithm $\mathscr{A}$.

For random variables $X$ and $Y$, write $\mathbb{E}^Y X = \mathbb{E}[X|Y]$ and $\mathbb{P}^Y[X]$ for the conditional expectation and (regular) conditional distribution, respectively, of $X$ given $Y$.[2] Besides the usual notions of KL divergence, mutual information, and conditional mutual information (see Appendix A for formal definitions), we rely on the following less common notion:

**Definition 1.1.** Let $X$, $Y$, and $Z$ be arbitrary random elements. Let $\otimes$ form product measures. The **disintegrated mutual information between $X$ and $Y$ given $Z$** is

$$I^Z(X;Y) = \mathrm{KL}(\mathbb{P}^Z[(X,Y)] \,\|\, \mathbb{P}^Z[X] \otimes \mathbb{P}^Z[Y]).$$

It follows immediately from definitions that $I(X,Y|Z) = \mathbb{E}I^Z(X,Y)$. Letting $\phi$ satisfy $\phi(Z) = I^Z(X;Y)$ a.s., define $I(X,Y|Z=z) = \phi(z)$. This notation is necessarily well defined only up to a null set under the marginal distribution of $Z$.

## 2 Methods

In this section, we establish generalization bounds for learning algorithms in terms of information-theoretic quantities (conditional mutual information, disintegrated mutual information, relative entropy) that depend on the unknown data distribution and the probabilistic properties of the learning algorithm. We then describe two complementary strategies that we employ to bound these otherwise intractable quantities. In Section 3, we apply these methods to the study of the Langevin algorithm and SGLD.

We make repeated use of generalized notions of *priors* and *posteriors*, which arise in the PAC-Bayes literature ([7, 21, 31], etc.) and relate to variational bounds on mutual information, which we will now describe: Consider learned parameters $W$, data $S$, and auxiliary variables $V$, viewed as random elements in $\mathscr{W}$, $Z^n$, etc., respectively. In PAC-Bayes, a generalized posterior is an arbitrary random measure on $\mathscr{W}$. In our setting, the **posterior, $Q$, (of $W$ given $S$ and $V$)** is the conditional distribution of $W$ given $S$ and $V$. (Formally, Q is a probability kernel, but one can think informally that $Q = f(S,V)$ for some measurable function taking values in the space of Borel probability measures, and so we will simply say that $Q$ is $\sigma(S,V)$-measurable.)

**Definition 2.1** (Data-dependent prior). Let $Q$ be a $\sigma(S,V)$-measurable posterior. A **(generalized) prior** P is a random measure on $\mathscr{W}$, measurable with respect to some sub-$\sigma$-algebra of $\sigma(S,V)$. A prior $P$ is said to be **data-dependent** if it is not independent of $S$.

Let $P$ be a $\mathscr{F}$-measurable data-dependent prior, where $\sigma(V) \subset \mathscr{F}$. Using a variational characterization of mutual information (see Appendix B.1), we have

$$\mathbb{E}^{\mathscr{F}}[\mathrm{KL}(Q \| P)] \geq I^{\mathscr{F}}(W;S) \text{ a.s.,} \tag{1}$$

with equality for $P = \mathbb{P}^{\mathscr{F}}[W]$. Therefore, if the expected KL divergence is small, $W$ contains little information about $S$ beyond what is already captured by $\mathscr{F}$. If the special case where the disintegrated mutual information is zero, then $W$ is independent of $S$ given $\mathscr{F}$. In the context of generalization, this implies that the data $S$ not contained in $\mathscr{F}$ can be used to form an unbiased estimate of the risk of $W$. The bounds we present below extend this logic to nonzero mutual information.

The utility of using data-dependent priors to control disintegrated mutual information depends on the balance of two effects: On the one hand, $I(W;S) \leq I(W;S|\mathscr{F})$, and so conditioning never improves a theoretical bound and may make it looser. On the other hand, $I(W;S)$ depends on the *unknown* data distribution and so distribution-independent bounds will often be very loose. In contrast, the KL divergence based on $P$ can exploit the information in $\mathscr{F} \subset \sigma(S,V)$ to obtain tighter data-dependent bounds on $I^{\mathscr{F}}(W;S)$.

In order to construct data-dependent priors, we partition the dataset $S$ in two halves, based on a random subset $J \subset \{1, \ldots, n\}$ with $\#J = m$ nonrandom. Let $J = \{j_1, \ldots, j_m\}$, The first half, $S_J = (Z_{j_1}, \ldots, Z_{j_m})$, contains $m$ points, which we will use to construct a data-dependent prior $P$. The second half, $S_J^c$, containing the remaining $n - m$ points, is independent of $P$. (Note that $S_J$ and $S_J^c$ are independent of $J$, since $m$ is nonrandom.)

This particular construction of data-dependent priors allow us to leverage a type of *non-uniform KL-stability*: the prior $P$ may exploit $S_J$ to make a data-dependent forecast of $Q$, yielding a bound, $B$, on the conditional expected generalization error (with respect to the remaining $n - m$ data points in $S_J^c$). Averaging over $S_J$, we obtain a bound on the (unconditional) expected generalization error.

**Definition 2.2.** Let $S_J, S_J^c$ be defined as above. Suppose that $\mathscr{F}$ is a $\sigma$-field with $\sigma(S_J) \subset \mathscr{F} \perp\!\!\!\perp \sigma(S_J^c)$. An expected generalization error bound based on a **data-dependent estimate** is one of the form

$$\mathbb{E}\left[R_{\mathscr{D}}(W) - \hat{R}_S(W)\right] \leq \mathbb{E}[B], \tag{2}$$

where $B$ is $\mathscr{F}$ measurable, and satisfies $\mathbb{E}^{\mathscr{F}}\left[R_{\mathscr{D}}(W) - \hat{R}_{S_J^c}(W)\right] \leq B$.

The idea of using data-dependent priors to obtain tighter bounds is standard in the PAC-Bayes literature [2, 10, 23, 28], but its utility in the present work is brought through by our introduction of data-dependent estimates. In the following section, we derive information-theoretic bounds on expected generalization error that can exploit data-dependent priors to form data-dependent estimates. We will then use these tools to study SGLD, without mixing assumptions.

## 2.1 Information-Theoretic Generalization Bounds based on Random Subsets of Data

Existing work by Xu and Raginsky [35] bounds the expected generalization error of a learning algorithm in terms of the mutual information between the random parameters and the data. The following result is a simple extension of [35, Thm. 1] that bounds the expected generalization error in terms of the mutual information between the parameters and a random subset of the data.

**Theorem 2.3** (Data-Dependent Mutual Information Bound). *Let $W$ be a random element in $\mathcal{W}$, let $S \sim \mathcal{D}^n$, and let $J \subseteq [n]$, $|J| = m$, be uniformly distributed and independent from $S$ and $W$. Suppose that $\ell(Z, w)$ is $\sigma$-subgaussian when $Z \sim \mathcal{D}$, for each $w \in \mathcal{W}$. Let $Q = \mathbb{P}^S[W]$, and let $P$ be a $\sigma(S_J)$-measurable data-dependent prior on $\mathcal{W}$. Then*

$$\mathbb{E}\left[R_{\mathcal{D}}(W) - \hat{R}_S(W)\right] \leq \sqrt{2\frac{\sigma^2}{n-m}I(W; S_J^c)} \leq \sqrt{2\frac{\sigma^2}{n-m}\mathbb{E}[\mathrm{KL}(Q\|P)]}.$$

The proof of this result can be found in Appendix B. When $m = 0$, this recovers [35, Thm. 1].

When the size of the subset is $m = n - 1$, this bound is weaker than [6, Prop. 1], due to the order of the concave square-root function and the expectation over the choice datapoint to be left out. This difference is addressed by our next result.

Randomization is one way that learning algorithms can control the mutual information between (a random subsets of) the data and the learned parameter. Let $U$ be a random element independent from $S$ and $J$, representing some aspect of the source of randomness used by the learning algorithm. Because $S \perp\!\!\!\perp \{J, U\}$ and $S \sim \mathcal{D}^n$, we have $(S_J, U) \perp\!\!\!\perp S_J^c$ and thus

$$I(W; S_J^c) \leq I(W; S_J^c | S_J, U) = \mathbb{E}I^{S_J,U}(W; S_J^c),$$

where the last equality follows from the definition of conditional mutual information. The next result shows that we can pull the expectation over both $S_J$ and $U$ outside the concave square-root function. In the case of SGLD, $U$ will be the sequence of minibatch index sets.

**Theorem 2.4** (Data-Dependent Disintegrated Mutual Information Bound). *Let $W$, $S$, and $J$ be as in Theorem 2.3, and let $U$ be independent from $S$ and $J$. Suppose that $\ell(Z, w)$ is $\sigma$-subgaussian when $Z \sim \mathcal{D}$, for each $w \in \mathcal{W}$. Let $Q = \mathbb{P}^{S,U}[W]$ and let $P$ be a $\sigma(S_J, U)$-measurable data-dependent prior on $\mathcal{W}$. Then*

$$\mathbb{E}\left[R_{\mathcal{D}}(W) - \hat{R}_S(W)\right] \leq \mathbb{E}\sqrt{2\frac{\sigma^2}{n-m}I^{S_J,U}(W; S_J^c)} \leq \mathbb{E}\sqrt{2\frac{\sigma^2}{n-m}\mathbb{E}^{S_J,U}\mathrm{KL}(Q\|P)}$$

The proof of this result can be found in Appendix B. Since $I^{S_J,U}(W; S_J^c)$ is $(S_J, U)$-measurable, we may use $S_J$ and $U$ to obtain a data-dependent bound. In the case that $m = n - 1$, our bound is similar to, but not strictly comparable to, [6, Prop. 1]. Our bound is incomparable due to our use of disintegrated mutual information, $I^{S_J}(W; S_J^c)$ and the fact that we take the expectations over the dataset outside of the convex square-root function. The disintegrated mutual information cannot be upper bounded by the full mutual information, $I(W, S_J^c)$, which appears in [6] (even by taking expectations under the square root using Jensen's inequality). However, Theorem 2.4 is essentially a disintegrated version of [6, Prop. 1]. In their actual SGLD expected generalization error bound, [6] controls the unconditional mutual information using the Lipschitz constant of the surrogate loss. Hence, one could easily recover the same bound using our result. The conditioning we have done, however, allows us to control the mutual information more carefully in order to achieve a tighter bound for SGLD than is provided by [6].

These bounds allow for a tradeoff: for large $m$, the mutual information is measured between the parameter and a small random subset of the data, and so we expect the mutual information to be small. (Indeed, this term will decrease monotonically in $m$.) At the same time, the $\frac{1}{n-m}$ term is larger, reflecting the reduced effect of averaging over only $n - m$ data to form our estimate of the empirical risk. It is unclear without further context whether this bound is tighter in the regime of small, intermediate, and large $m$. In fact, we find that, for the bounds we derive in our applications, $m = n - 1$ is optimal. This difference materially affects the quality and tightness of the bounds, as is discussed in Remark 3.4. However, for $m = n - 1$ and bounded loss, the following bound is tighter, while it is incomparable for other values of $m$.

**Theorem 2.5** (Data-Dependent KL Bound). *Let W, S, J, and U be as in Theorem 2.4. Let $Q = \mathbb{P}^{S,U}[W]$ and let P be a $\sigma(S_J, U)$-measurable data-dependent prior on $\mathcal{W}$. Suppose that $\ell(Z, w)$ is $[a_1, a_2]$-bounded a.s. when $Z \sim \mathcal{D}$, for each $w \in \mathcal{W}$.*

$$\mathbb{E}\left[R_{\mathcal{D}}(W) - \hat{R}_S(W)\right] \leq \mathbb{E}\sqrt{\frac{(a_2 - a_1)^2}{2} \operatorname{KL}(Q \| P)}.$$

The proof of this result can be found in Appendix B. For an analytic comparison of the three bounds in the case that $m = n - 1$, see Appendix F. Remark B.2 explains why this result is only stated for bounded loss functions.

## 2.2 Decomposing KL Divergences and Mutual Information for Sequential Algorithms

Consider an iterative learning algorithm, and let $W_0, W_1, W_2, \ldots W_T \in \mathcal{W}$ be the parameters during the course of $T$ iterations. In light of the variational bound for mutual information, we can obtain a generalization bound for $W_T$ by bounding the expected KL divergences between the conditional distribution $\mathbb{P}^{S_J}[W_T]$ and some $S_J$-measurable "prior" distribution $P(Z)$. Unfortunately, the first distribution has no known tractable representation. Pensia, Jog, and Loh [24] use monotonicity to bound a mutual information involving the terminal parameter with one involving the full trajectory, then use the chain rule to decompose this into a sum of conditional mutual informations. The same principles allow us to first bound the terminal KL divergence by the KL for the full trajectory, and then decompose the KL divergence for the full trajectory over each individual step.

Setting some notation, let $T$ be a nonnegative integer, let $[T]_0 = \{0, 1, 2, \ldots, T\}$, let $\mu$ be a distribution on $\mathcal{W}^{[T]_0}$, and let $X$ be a random variable with distribution $\mu$. We are interested in naming certain marginal and conditional distributions (disintegrations) related to $\mu$. In particular, for $t \in [T]_0$, let

  i) $\mu_t = \mathbb{P}[X_t]$, the marginal law of $X_t$;
  ii) $\mu_{t|} = \mathbb{P}^{X_{0:(t-1)}}[X_t]$, the conditional law of $X_t$ given $X_{0:(t-1)}$; and
  iii) $\mu_{0:t} = \mathbb{P}[X_{0:t}]$, the marginal law of $X_{0:t}$.

**Proposition 2.6** (Decomposition of KL Divergences). *Let $Q, P$ be probability measures on $\mathcal{W}^{[T]_0}$. Suppose that $Q_0 = P_0$. Then*
$$\operatorname{KL}(Q_T \| P_T) \leq \operatorname{KL}(Q \| P) = \sum_{t=1}^{T} \mathbb{E}_{Q_{0:(t-1)}}[\operatorname{KL}(Q_{t|} \| P_{t|})].$$

*where, as per Section 1.2, $Q_{t|}$ is the conditional law of t-th iterate given the previous iterates, and so $\operatorname{KL}(Q_{t|} \| P_{t|})$ is a random variable which depends the $(W_0, \ldots W_{t-1}) \sim Q_{0:t-1}$.*

The proof of this result may be found in Appendix B.

Considering the KL between full trajectories may yield a loose upper bound on the KL between terminal parameters (in particular, when the trajectory cannot be inferred from the terminus). We gain, however, analytical tractability, as we will see in the next section when we analyze particular algorithms stepwise. In fact, many bounds that appear in the literature implicitly require this form of incrementation. Our approach based on the KL divergence and data-dependent priors gives us much tighter control of the KL divergence contribution of each step.

# 3 Generalization Bounds for Specific Algorithms

Now that we have all of the theoretical tools required, we may establish bounds on the generalization error of specific noisy iterative learning algorithms by inventing sensible data-dependent priors. The use of a data-dependent prior which closely forecasts the true algorithm in each step is key in establishing tighter generalization bounds. We first consider the stochastic gradient Langevin dynamics (SGLD) algorithm [34], then handle its full batch counterpart the (unadjusted) Langevin algorithm [9, 11], which we will refer to informally as Langevin dynamics (LD). Note that the loss and risk functions used for training, $(\tilde{\ell}, \tilde{R}_{\mathcal{D}}, \tilde{R}_S)$, need not be the same loss functions used for assessing performance and generalization error, $(\ell, R_{\mathcal{D}}, \hat{R}_S)$, as explained in Section 1.2.

## 3.1 Stochastic Gradient Langevin Dynamics

Let $\eta_t$ to be the learning rate at time $t$; $\beta_t$ be the inverse temperature at time $t$; and $\varepsilon_t$, i.i.d. $\mathcal{N}(0, \mathbb{I}_d)$. Let $b_t$ be the minibatch size at time $t$. We are interested in stochastic gradient Langevin dynamics,

whose iterates are given by

$$W_{t+1} = W_t - \eta_t \nabla \tilde{R}_{S_t}(W_t) + \sqrt{2\eta_t/\beta_t}\, \varepsilon_t. \tag{3}$$

where $\tilde{R}_{S_t}(w) = \frac{1}{b_t} \sum_{z \in S_t} \tilde{\ell}(w, z)$, and $S_t$ is a subset of $S$ of size $b_t$ sampled uniformly at random with a sampling procedure which is independent of $S$, and independent of $\{\varepsilon_t\}_{t \geq 0}$. The $b_t$ data points in $S_t$ are chosen *without replacement*.

### 3.1.1 A data-dependent prior for SGLD

Let $S_J$ be a random subset of $S$, of size $m$, chosen independently from $W_0, W_1, \ldots$, and independently of the sequence of minibatches, $\{S_t\}_{t \geq 0}$. Let the set of indices appearing in the $t$-th minibatch be denoted by $K_t$, so that $S_t = S_{K_t}$ for each $t$. By assumption, each $K_t$ is a uniformly random subset of $\{1, \ldots, n\}$ of size $b_t$. We set $U = (K_1, \ldots K_T)$, as to match the notation in the theorems of Section 2.1. Let $S_{J_t} = S_J \cap S_t = S_{J \cap K_t}$ and let $b_t' = \#S_{J_t}$. Let $S_t^c = S_t \setminus S_J = S_{K_t \setminus J}$ and $b_t^c = b_t - b_t'$. Define

$$\xi_t = \frac{b_t^c}{b_t} \left( \nabla \tilde{R}_{S_t^c}(W_t) - \nabla \tilde{R}_{S_J}(W_t) \right). \tag{4}$$

Let $Q(S, U)$ be the joint law of $(W_0, \ldots, W_T)$ given a dataset $S$ and minibatch sequence $U$. Then $Q(S, U)$ is a random measure as it depends on the random dataset $S$ and the sequence of indices $U$. It follows from Eq. (3) that $Q(S, U)_{t|}$ is multivariate normal with mean $\mu_{Q,t}(S, U) = W_t - \eta_t \nabla \tilde{R}_S(W_t)$ and covariance $2\frac{\eta_t}{\beta_t} \mathbb{I}_d$. Consider the data-dependent prior defined so that its conditional $P_{t|}(S_J, U)$ is a multivariate normal with covariance $2\frac{\eta_t}{\beta} \mathbb{I}_d$, and with mean

$$\mu_{P,t}(S_J, U) = W_t - \eta_t \left( \frac{b_t'}{b_t} \nabla \tilde{R}_{S_{J_t}}(W_t) + \frac{b_t - b_t'}{b_t} \nabla \tilde{R}_{S_J}(W_t) \right).$$

Note that $\mu_{Q,t}(S, U) - \mu_{P,t}(S_J, U) = \eta_t \xi_t(S, \text{idx})$. Thus the one-step KL divergence satisfies

$$2\text{KL}(Q_{t+1|}(S, \text{idx}) \| P_{t+1|}(S_J, U)) = \frac{\beta_t \eta_t}{4} \|\xi_t\|_2^2$$

Applying Proposition 2.6, we have (almost surely over the choice of $(S, J, U)$)

$$2\text{KL}(Q_T(S, U) \| P_T(S_J, U)) \leq \sum_{t=1}^T \mathbb{E}^{S,J,U} \text{KL}(Q_{t|}(S, U) \| P_{t|}(S_J, U)) = \sum_{t=1}^T \mathbb{E}^{S,J,U} \frac{\beta_t \eta_t}{4} \|\xi_t\|_2^2.$$

Note that $\xi_t$ depends on the exact weight sequence, and hence is $\sigma(S, J, U, W_{t-1})$-measurable, but not $\sigma(S, J, U)$-measurable. Hence, $\mathbb{E}^{S,J,U} \frac{\beta_t \eta_t}{8} \|\xi_t\|_2^2$ is a $\sigma(S, J, U)$-measurable for each $t$.

### 3.1.2 Expected Generalization Error Bounds for SGLD

**Theorem 3.1** (Expected Generalization Error Bounds for SGLD). *Let $\{W_t\}_{t \in [T]}$ denote the iterates of SGLD. Let the batch size be constant, $b_t = b$. If $\ell(Z, w)$ is $\sigma$-subgaussian for each $w \in \mathcal{W}$, then*

$$\mathbb{E}(R_{\mathscr{D}}(W_T) - R_S(W_T)) \leq \mathbb{E}\sqrt{\frac{\sigma^2}{n-m} \sum_{t=1}^T \frac{\beta_t \eta_t}{4} \mathbb{E}^{S,J,U} \|\xi_t\|_2^2} \leq \frac{\sigma}{2} \sqrt{\frac{n}{(n-1)^2} \sum_{t=1}^T \left( \frac{1}{b} + \frac{1}{n} \frac{n-m-1}{m} \right) \beta_t \eta_t \text{tr}(\mathbb{E}[\hat{\Sigma}_t(S)])}$$

(5)

*and if $\ell(Z, w)$ is $[a_1, a_2]$-bounded, and if $m = n - 1$, then*

$$\mathbb{E}(R_{\mathscr{D}}(W_T) - R_S(W_T)) \leq \mathbb{E}\sqrt{\frac{(a_2 - a_1)^2}{4} \sum_{t=1}^T \frac{\beta_t \eta_t}{4} \mathbb{E}^{S,J,U} \|\xi_t\|_2^2} \leq \left[ \frac{(a_2 - a_1)^2 n}{4(n-1)^2 b} \right]^{1/2} \mathbb{E}\sqrt{\sum_{t=1}^T \frac{\beta_t \eta_t}{4} \text{tr}(\mathbb{E}^S[\hat{\Sigma}_t(S)])}$$

(6)

*where $\hat{\Sigma}_t(S) = \underset{\substack{Z \sim \text{Unif}(S)}}{Var^{W_t, S}} (\nabla \tilde{R}_Z(W_t))$ is the finite population variance matrix of surrogate gradients.*

*Proof.* The results are the direct combinations of Theorem 2.4 and Propositions 2.6 and B.1; and Theorem 2.5 and Proposition 2.6, respectively, with our data-dependent prior. Jensen's inequality is used to move expectations under $\sqrt{\cdot}$. Lemma D.2 expresses the results in terms of $\hat{\Sigma}$. $\qquad \square$

*Remark* 3.2. Suppose that $\beta_t = \beta$, $b_t = b$, and $m = n-1$. Under uniform moment conditions on $\mathbb{E}^{S_J,J,U}\|\xi_t\|_2^2$, our generalization error bounds in Eq. (5) is clearly $O\big(\sqrt{(\beta/bn)\sum_{t\leq T}\eta_t}\big)$. Since $\xi_t = 0$ whenever $K_t \subset J$, we find that our first bound in Eq. (5) is also $O\big((1/n)\sum_{t\leq T}\sqrt{\beta\eta_t}\big)$. To see this, notice that for non-negative random variables $C_t$ and $B_t \sim \text{Ber}(p)$,

$$\mathbb{E}\sqrt{\sum_{t=1}^T B_t C_t} \leq \mathbb{E}[\sum_{t=1}^T B_t \sqrt{C_t}] = p\sum_{t=1}^T \mathbb{E}[\sqrt{C_t}|B_t = 1].$$

When $m = n-1$, taking $B_t = I_{\xi_t \neq 0}$, $p = b/n$, $C_t = \frac{\beta_t\eta_t}{8}\mathbb{E}^{S_J,J,U}\|\xi_t\|_2^2$ yields the stated rate. ◁

### 3.2 Langevin Dynamics

Under the same notation as above, the iterates of the Langevin dynamics algorithm are given by

$$W_{t+1} = W_t - \eta_t \nabla\tilde{R}_S(W_t) + \sqrt{2\eta_t/\beta_t}\,\varepsilon_t. \tag{7}$$

#### 3.2.1 Expected Generalization Error Bounds for LD

We can recover bounds generalization error bounds for LD as a special case of SGLD when the batch size is the dataset size, $b_t = n$ for all $t$. The data-dependent prior is the same as for SGLD.

**Theorem 3.3** (Expected Generalization Error Bounds for Langevin Dynamics). *Let $\{W_t\}_{t\in[T]}$ denote the iterates of the Langevin dynamics algorithm. If $\ell(Z,w)$ is $\sigma$-subgaussian for each $w \in \mathscr{W}$, then*

$$\mathbb{E}(R_{\mathscr{D}}(W_T) - R_S(W_T))4 \leq \sqrt{\frac{\sigma^2}{(n-1)m}\sum_{t=1}^T \frac{\beta_t\eta_t}{4}\mathbb{E}\text{tr}(\hat{\Sigma}_t(S))}, \tag{8}$$

*and if $\ell(Z,w)$ is $[a_1,a_2]$-bounded and $m = n-1$, then*

$$\mathbb{E}(R_{\mathscr{D}}(W_T) - R_S(W_T)) \leq \mathbb{E}\sqrt{\frac{(a_2-a_1)^2}{4}\sum_{t=1}^T \frac{\beta_t\eta_t}{4}\mathbb{E}^{S_J}\|\xi_t\|_2^2} \leq \frac{a_2-a_1}{2(n-1)}\mathbb{E}\sqrt{\sum_{t=1}^T \frac{\beta_t\eta_t}{4}\mathbb{E}^S\text{tr}(\hat{\Sigma}_t(S))},$$

*where $\hat{\Sigma}_t(S) = \underset{Z\sim Unif(S)}{Var^{W_t,S}}(\nabla\tilde{R}_Z(W_t))$ is the finite population variance matrix of surrogate gradients.*

For asymptotic properties of this bound when $\tilde{\ell}$ is $L$-Lipschitz, as in [24], see Appendix E. For a simple analytic worked example of mean estimation using Langevin dynamics, refer to Appendix G.

*Remark* 3.4 (Dependence of our bounds on the subset size, $m$). The choice of $m \in \{1,\ldots,n\}$ can make a material difference in the quality of the bound and whether it is vacuous or not. As seen in Eq. (8), if $m$ is $\Omega(n)$ then the upper bound on expected generalization error is $O(\beta/n)$. If $\beta$ is $\Omega(\sqrt{n})$, as is typical in practice, then overall, the bound is $O(n^{-1/2})$. If, on the other hand, $m$ is $o(n)$ then the order of the bound with respect to $n$ would be lower—in particular if $m$ is $O(\sqrt{n})$ then our bound would not be decreasing in $n$ for $\beta$ of order $\Omega(\sqrt{n})$. ◁

## 4 Empirical Results

We have developed bounds that depend on the gradient prediction residual of our data dependent priors (which we call the *incoherence* of the gradients), rather than on the gradient norms (as in [22]) or Lipschitz constants (as in [6, 24]). The extent to which this represents an advance is, however, an empirical question. The functional form of our bounds and those in the cited work are nearly identical. The first key differences between our work and others is the replacement of gradient norms ($\|\nabla\tilde{R}_t\|^2$) and Lipschitz constants in other work with gradient prediction residual, ($\|\xi_t\|$) in our work. The second key difference is the order of expectations and square-roots, which favor our bounds due to Jensen's inequality. In this section, we perform an empirical comparison of the gradient prediction residual of our data dependent priors and the gradient norm across various architectures and datasets. This illustrates the first of the differences, the quantities appearing in the bound. Our results indicate that that our data-dependent priors yield significantly tighter results, as the sum of square gradient incoherences of our data dependent priors are between $10^2$ and $10^4$ times smaller than the sum of square gradient norms in the experiments we ran.

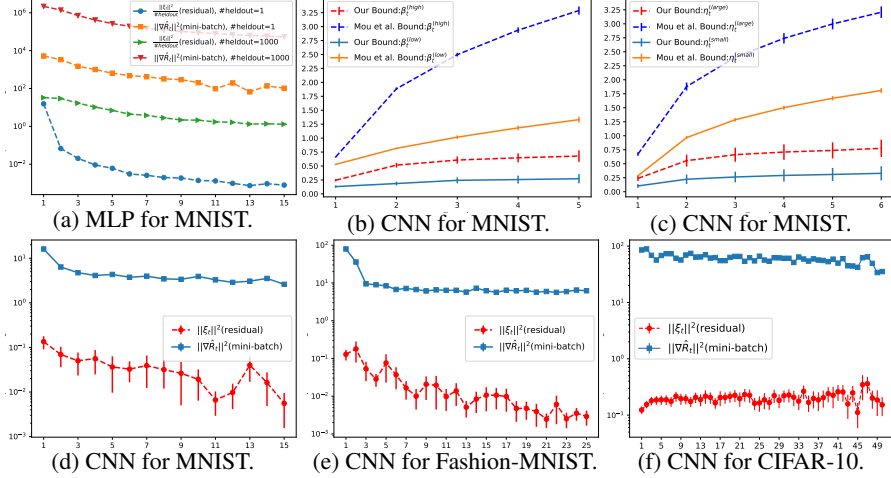

**Figure 1:** Numerical results for various datasets and architectures. All $x$-axes show the number of Epochs of training. Fig. 1a shows the effect of different amounts of heldout data on the summands appearing in our bound, and what those would be if we upper bounded the *incoherence* $\|\xi\|$ by $\|\nabla\hat{R}\|$ when it is not 0. Fig. 1b compares a Monte Carlo estimate of our bound with that of [22] and shows the effect of inverse temperature on each. Fig. 1c compares a Monte Carlo estimate of our bound with that of [22] and shows the effect of learning rate on each. Figs. 1d to 1f compare the summands appearing in our bound and those of [22] across datasets.

In Fig. 1, we compare $\|\xi_t\|^2$ and $\|\nabla\tilde{R}_t\|^2$ in order to assess the improvement our methods bring over existing results for SGLD. Specifically, the values of each plot are the averages of $\sqrt{\eta\beta}\|\xi_t\|/b$ and $\sqrt{\eta\beta}\|\nabla\tilde{R}_{S_t}\|/b$ over an epoch. These serve as estimates of the per-epoch contributions to the respective summations in our Theorem 3.1 and the bound of Mou et al. (Thm. 2 therein, when there is no $L_2$-regularization). The average and standard error of both expressions taken over multiple runs are displayed. Bounds from related work that depend on Lipschitz constants would further upper bound what we show for [22], by replacing $\|\nabla\tilde{R}_t\|$ with a Lipschitz constant. The Lipschitz constant could be lower bounded by the largest observed gradient norm, and would be off the chart.

From Fig. 1a, we see that the empirical performance reflects our analytical results that the bound is tighter for large $m$. As can be inferred from Eq. (4), the difference between $\|\xi_t\|^2$ and $\|\nabla\tilde{R}_t\|^2$ increases with $m$. From Figs. 1d to 1f we see that the squared gradient incoherence, $\|\xi_t\|^2$, are between 100 and 10,000 times smaller than the squared gradient norms, $\|\nabla\tilde{R}\|^2$ in all of these examples.

Using Monte Carlo simulation, we compared estimates of our expected generalization error bounds with (coupled) estimates of the bound from [22]. The results, in Figs. 1b and 1c, show that our bounds are materially tighter, and remain non-vacuous after many more epochs. Fig. 1b also compares the two generalization error bounds for different inverse temperature schedules. Fig. 1c compares the two generalization error bounds based for different learning rate schedules. It can inferred from Figs. 1b and 1c that our proposed bound yields to tighter values when the learning rate and the inverse temperature are small. However, it should be noted that with small learning rate and the inverse temperature, it would be difficult to have a very low training error when the empirical risk minimization is performed using SGLD.

The details of our model architectures, temperature, learning rate schedules and hyperparameter selections may be found in Appendix H. We did not aim to achieve the state-of-the art predictive performance. With further tuning, the prediction results could be improved.

### Acknowledgments

JN is supported by an NSERC Vanier Canada Graduate Scholarship, and by the Vector Institute. MH was supported by a MITACS Accelerate Fellowship with Element AI. DMR is supported by an NSERC Discovery Grant and an Ontario Early Researcher Award. This research was carried out in part while GKD and DMR were visiting the Simons Institute for the Theory of Computing.

## Footnotes

[2]We fix arbitrary versions and assume regular versions of conditional distributions exist.

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
