[Supplementary Material]

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

# A Common Definitions

In this appendix, we collect together a few standard definitions from information theory. Let $P, Q$ be probability measures on a common measurable space. Write $Q \ll P$ when $Q$ is absolutely continuous with respect to $P$, i.e., for all measurable subsets $A$, $Q(A) = 0$ if $P(A) = 0$. By the Radon–Nikodym theorem, when $Q \ll P$, there exists a measurable function $\frac{dQ}{dP}$, called a Radon–Nikodym derivative or density, such that $Q(A) = \int_A \frac{dQ}{dP} dP$ for all measurable subsets $A$. The **KL divergence** (or **relative entropy**) **of** $Q$ **with respect to** $P$, written $\mathrm{KL}(Q \| P)$, is defined to be $\int \log \frac{dQ}{dP} dQ$ when $Q \ll P$ and is defined to be infinity otherwise.

Given random elements $X$ and $Y$, the **mutual information between** $X$ **and** $Y$, written $I(X; Y)$ is

$$I(X; Y) = \mathrm{KL}(\mathbb{P}[(X, Y)] \| \mathbb{P}[X] \otimes \mathbb{P}[Y]),$$

where $\otimes$ forms the product measure. Given another random element $Z$, the **conditional mutual information between** $X$ **and** $Y$ **given** $Z$ is defined to be $I(X; Y | Z) = I(X; (Y, Z)) - I(X; Z) = I((X, Z); Y) - I(Z; Y)$.

Relative entropy and mutual information satisfy many well-known properties: For example, relative entropy and mutual information are nonnegative; $X \perp\!\!\!\perp Y \iff I(X; Y) = 0$; and $I(X; Y) \leq I(X; (Y, Z))$. From this last inequality, one may deduce that $I(X; Y) \leq I(X; Y | Z)$ when $X \perp\!\!\!\perp Z$.

# B Proofs of Results

## B.1 Bounding Mutual Information by KL Divergence

The following is a well-known result that allows one to bound mutual information by the expectation of the KL divergence of a "posterior" with respect to a "prior" (where these terms are taken to have their more general interpretation from PAC-Bayesian theory, as opposed to the classical Bayesian theory).

**Proposition B.1** (Variational Representation of Mutual Information). *Let $X$ and $Y$ be random elements. Then, for all probability measures $P$ on the same space as $Y$,*

$$I(X; Y) \leq \mathbb{E}[\mathrm{KL}(\mathbb{P}^X[Y] \| P)],$$

*with equality for $P = \mathbb{E}[\mathbb{P}^X[Y]] = \mathbb{P}[Y]$.*

The result is implicit in [18] and is considered folklore in the literature (e.g., it is referenced without proof in [7]). For a simple derivation, see [25, Eq. (1)]. Given another random element $Z$, it follows immediately by the disintegration theorem [17, Thm. 6.4] that, for all $Z$-measurable random probability measures $P$ on the same space as $Y$,

$$I^Z(X; Y) \leq \mathbb{E}^Z[\mathrm{KL}(\mathbb{P}^{X,Z}[Y] \| P)] \text{ a.s.},$$

with a.s. equality for $P = \mathbb{E}^Z[\mathbb{P}^{X,Z}[Y]] = \mathbb{P}^Z[Y]$.

## B.2 Proofs of Main Results

*Proof of Theorem 2.3.* Let $\tilde{W}$ be a random element in $\mathscr{W}$ such that $W \overset{d}{=} \tilde{W}$ and $\tilde{W} \perp\!\!\!\perp S_J^c$. Let $\mathscr{G}$ denote the class of all functions $g$ such that $\mathbb{E} \exp(g(\tilde{W}, S_J^c)) < \infty$. Then

$$I(W; S_J^c) = \mathrm{KL}(\mathbb{P}(W, S_J^c) \| \mathbb{P}(\tilde{W}, S_J^c)) \tag{9}$$

$$= \sup_{g \in \mathscr{G}} \mathbb{E} g(W, S_J^c) - \log \mathbb{E} e^{g(\tilde{W}, S_J^c)} \tag{10}$$

where the second equality follows from the Donsker–Varadhan variational formula [5, Prop. 4.15] (see also [8]). Let $f(w, s) = R_{\mathscr{D}}(w) - \hat{R}_s(w)$ so that $\mathbb{E} f(W, S_J^c) = \mathbb{E} R_{\mathscr{D}}(W) - \mathbb{E} \hat{R}_{S_J^c}(W)$ and $\mathbb{E} f(\tilde{W}, S_J^c) = 0$. Let $\psi$ be the cumulant generating function of $f(\tilde{W}, S_J^c)$ and let $D$ be the domain on which this cumulant generating function is defined. Then $\lambda f \in \mathscr{G}$ exactly when $\lambda \in D$. Then, for

every $\lambda \in D$,

$$\sup_{g \in \mathscr{G}} \mathbb{E}g(W, S_J^c) - \log \mathbb{E}e^{g(\tilde{W}, S_J^c)} \geq \lambda \mathbb{E}f(W, S_J^c) - \log \mathbb{E}e^{\lambda f(\tilde{W}, S_J^c)} \tag{11}$$

$$= \lambda \mathbb{E}\left[R_{\mathscr{D}}(W) - \hat{R}_{S_J^c}(W)\right] - \psi(\lambda). \tag{12}$$

By rearranging and optimizing over $\lambda$, we find that

$$\mathbb{E}\left[R_{\mathscr{D}}(W) - \hat{R}_{S_J^c}(W)\right] \leq \inf_{\lambda \in D} \frac{\psi(\lambda) + I(W; S_J^c)}{\lambda}.$$

Because the subset $J$ is random and independent of $(S, W)$, we have $\mathbb{E}\hat{R}_{S_J^c}(W) = \mathbb{E}\hat{R}_S(W)$. Hence,

$$\mathbb{E}\left[R_{\mathscr{D}}(W) - \hat{R}_S(W)\right] = \mathbb{E}\left[R_{\mathscr{D}}(W) - \hat{R}_{S_J^c}(W)\right] \leq \inf_{\lambda \in D}\left[\frac{\psi(\lambda) + I(W; S_J^c)}{\lambda}\right].$$

At this point we have established a slightly more abstract result that permits applications beyond the subgaussian case. By the subgaussian hypothesis, $f(w, S_J^c)$ is itself $\sigma_{n-m}$-subgaussian for each $w \in \mathscr{W}$, and so the bound above reduces to

$$\mathbb{E}\left[R_{\mathscr{D}}(W) - \hat{R}_S(W)\right] \leq \sqrt{2\sigma_{n-m}^2 I(W; S_J^c)}$$

using the same optimization argument as in [6], [35], etc. From the proof of Theorem C.1, $\sigma_{n-m} \leq \frac{\sigma}{\sqrt{n-m}}$, completing the proof. □

*Proof of Theorem 2.4.* Let $\tilde{W}$ be a random element in $\mathscr{W}$ such that $(W, S_J, U) \overset{\mathrm{d}}{=} (\tilde{W}, S_J, U)$ and $\tilde{W} \perp\!\!\!\perp S_J^c \mid \{S_J, U\}$. Let $Q$ and $P$ satisfy $Q(S_J, U) = \mathbb{P}^{S_J, U}[W, S_J^c]$ and $P(S_J, U) = \mathbb{P}^{S_J, U}[\tilde{W}, S_J^c]$ a.s. By the Donsker–Varadhan variational formula [5, Prop. 4.15] and the disintegration theorem [17, Thm. 6.4], with probability one, for all measurable functions $g$ such that $P(S_J, U)(\exp g) < \infty$,

$$I^{S_J, U}(W; S_J^c) = \mathrm{KL}(Q(S_J, U) \| P(S_J, U))$$
$$\leq Q(S_J, U)(g) - \log P(S_J, U)(\exp g).$$

Let $f(w, s) = R_{\mathscr{D}}(w) - \hat{R}_s(w)$. Note that, a.s., $P(S_J, U)(f) = \mathbb{E}^{S_J, U}[f(\tilde{W}, S_J^c)] = 0$ and

$$Q(S_J, U)(f) = \mathbb{E}^{S_J, U}[f(W, S_J^c)] = \mathbb{E}^{S_J, U}[R_{\mathscr{D}}(W) - \hat{R}_{S_J^c}(W)].$$

Let $\psi$ be the cumulant generating function of $P(S_J, U)$, i.e., $\psi(\lambda; S_J, U) = \log P(S_J, U)(\exp\{\lambda f\})$. Let $D(S_J, U) = \{\lambda \in \mathbb{R} : \psi(\lambda; S_J, U) < \infty\}$. Then, with probability one, for all $\lambda \in D(S_J, U)$,

$$I^{S_J, U}(W; S_J^c) \geq \lambda \mathbb{E}^{S_J, U}\left[R_{\mathscr{D}}(W) - \hat{R}_{S_J^c}(W)\right] - \psi(\lambda; S_J, U).$$

Rearranging, with probability one,

$$\mathbb{E}^{S_J, U}\left[R_{\mathscr{D}}(W) - \hat{R}_{S_J^c}(W)\right] \leq \inf_{\lambda \in D(S_J, U)} \frac{I^{S_J, U}(W; S_J^c) + \psi(\lambda; S_J, U)}{\lambda}.$$

Because $W \perp\!\!\!\perp J$ and the subset $J$ is random and uniformly distributed,

$$\mathbb{E}\left[R_{\mathscr{D}}(W) - \hat{R}_S(W)\right] = \mathbb{E}\,\mathbb{E}^{S_J, U}\left[R_{\mathscr{D}}(W) - \hat{R}_{S_J^c}(W)\right]$$

$$\leq \mathbb{E}\left[\inf_{\lambda \in D(S_J, U)} \frac{I^{S_J, U}(W; S_J^c) + \psi(\lambda; S_J, U)}{\lambda}\right].$$

At this point we have established a slightly more abstract result that permits applications beyond the subgaussian case. By the subgaussian hypothesis, $f(w, S_J^c)$ is itself $\sigma_{n-m}$-subgaussian for each $w \in \mathscr{W}$, and so the bound above reduces to

$$\mathbb{E}\left[R_{\mathscr{D}}(W) - \hat{R}_S(W)\right] \leq \mathbb{E}\sqrt{2\sigma_{n-m}^2 I^{S_J, U}(W; S_J^c)}$$

using the same optimization argument as in [6], [35], etc. From the proof of Theorem C.1, $\sigma_{n-m} \leq \frac{\sigma}{\sqrt{n-m}}$, completing the proof. □

*Proof of Theorem 2.5.* For any two random measures $P(S_J, U), Q(S, U)$, the Donsker–Varadhan variational formula [5, Prop. 4.15] and the disintegration theorem [17, Thm. 6.4], give that with probability one

$$\mathrm{KL}(Q(S,U) \| P(S_J, U)) \geq \sup_{g \in \mathscr{G}} (Q(S,U)(g) - P(S_J, U)(g) - \log [P(S_J, U) (\exp(g - P(S_J, U)(g)))]),$$

where $\mathscr{G}(S_J, U) = \{g : P(S_J, U)(\exp g) < \infty\}$.

Taking $g(w) = \lambda \left( R_{\mathscr{D}}(w) - \hat{R}_{S_J^c}(w) \right)$, and letting

$$R_{\mathscr{D}}(Q) = Q(S,U)(R_{\mathscr{D}}) \qquad\qquad R_{\mathscr{D}}(P) = P(S_J, U)(R_{\mathscr{D}})$$
$$\hat{R}_{S_J^c}(Q) = Q(S,U)(\hat{R}_{S_J^c}) \qquad\qquad \hat{R}_{S_J^c}(P) = P(S_J, U)(\hat{R}_{S_J^c})$$

where, for brevity, we have used the short hand $Q = Q(S,U)$ and $P = P(S_J, U)$. Then, with probability one

$$\mathrm{KL}(Q(S,U) \| P(S_J, U))$$
$$\geq \lambda \left( R_{\mathscr{D}}(Q) - \hat{R}_{S_J^c}(Q) - \left( R_{\mathscr{D}}(P) - \hat{R}_{S_J^c}(P) \right) \right)$$
$$- \log \left[ P(S_J, U) \left( \exp \left( \lambda \left( R_{\mathscr{D}} - \hat{R}_{S_J^c} - \left( R_{\mathscr{D}}(P) - \hat{R}_{S_J^c}(P) \right) \right) \right) \right) \right]$$

Let

$$\psi(\lambda; S, J, U) = \log \left[ P(S_J, U) \left( \exp \left( \lambda \left( R_{\mathscr{D}} - \hat{R}_{S_J^c} - \left( R_{\mathscr{D}}(P) - \hat{R}_{S_J^c}(P) \right) \right) \right) \right) \right],$$

and $D(S, J, U) = \{\lambda \in \mathbb{R} : \psi(\lambda; S, J, U) < \infty\}$. With probability one

$$\left( R_{\mathscr{D}}(Q) - \hat{R}_{S_J^c}(Q) - \left( R_{\mathscr{D}}(P) - \hat{R}_{S_J^c}(P) \right) \right) \leq \inf_{\lambda \in D(S,J,U)} \frac{\mathrm{KL}(Q(S,U) \| P(S_J, U)) + \psi(\lambda; S, J, U)}{\lambda}$$

Since $P(S_J, U)$ is independent of $S_J^c$ then we have $\mathbb{E}^{S_J, J, U} \left[ R_{\mathscr{D}}(P) - \hat{R}_{S_J^c}(P) \right] = 0$. Hence, by averaging over $S_J^c$ (equivalently, taking the conditional expectation conditional on $(S_J, J, U)$) we have, with probability one

$$\mathbb{E}^{S_J, J, U} \left[ R_{\mathscr{D}}(Q) - \hat{R}_{S_J^c}(Q) \right] = \mathbb{E}^{S_J, J, U} \left[ R_{\mathscr{D}}(Q) - \hat{R}_{S_J^c}(Q) - \left( R_{\mathscr{D}}(P) - \hat{R}_{S_J^c}(P) \right) \right]$$
$$\leq \mathbb{E}^{S_J, J, U} \left[ \inf_{\lambda \in D(S,J,U)} \frac{\mathrm{KL}(Q(S,U) \| P(S_J, U)) + \psi(\lambda; S, J, U)}{\lambda} \right]$$

Finally, by taking the full expectation, since $J \perp\!\!\!\perp Q(S,U)$ we get:

$$\mathbb{E} \left[ R_{\mathscr{D}}(Q(S,U)) - \hat{R}_S(Q(S,U)) \right] \leq \mathbb{E} \left[ \inf_{\lambda > 0} \frac{\mathrm{KL}(Q(S,U) \| P(S_J, U)) + \psi_{S,J,U}(\lambda)}{\lambda} \right]$$

where the final $\mathrm{KL}(Q(S,U) \| P(S_J, U))$ on the right hand side is between two random measures, and hence is a random variable depending on $(S, J, U)$; and the expectation on the right hand side integrates over $(S, J, U)$.

If, for $(V \mid S_J, U) \sim P(S_J, U)$ it is the case that $\left( R_{\mathscr{D}}(V) - \hat{R}_{S_J^c}(V) \right)$ is $\sigma$-subgaussian for any $(S, J, U)$, then this can be optimized to get

$$\mathbb{E} \left[ R_{\mathscr{D}}(Q(S,U)) - \hat{R}_S(Q(S,U)) \right] \leq \mathbb{E} \sqrt{2\sigma^2 \, \mathrm{KL}(Q(S,U) \| P(S_J, U))}$$

When the loss is $[a_1, a_2]$-bounded then $R_{\mathscr{D}}(V) - \hat{R}_{S_J^c}(V)$ is $\frac{a_2 - a_1}{2}$ subgaussian which completes the proof.

$\square$

*Remark* B.2 (Why does Theorem 2.5 use a boundedness assumption instead of a subgaussian assumption?). Note that we needed the boundedness assumption because even if, for $Z \sim \mathscr{D}$, $\ell(Z,w)$ was subgaussian (uniformly in $w \in \mathscr{W}$) it may not be the case that for $(V \mid S_J,U) \sim P(S_J,U)$, $\left( R_{\mathscr{D}}(V) - \hat{R}_{S_J^c}(V) \right)$ is subgaussian. In contrast, in the proofs of Theorem 2.3, Theorem 2.4, and Theorem C.1 the expectations over $S_J^c$ included in the definition of the required cumulant generating functions let us take advantage of the subgaussian property of $\ell(Z,w)$. ◁

*Proof of Proposition 2.6.*

$$\text{KL}(Q_T \| P_T) \leq \text{KL}(Q_T \| P_T) + \mathbb{E}\text{KL}(Q_{|T} \| P_{|T}) = \text{KL}(Q \| P).$$

This tells us that the KL divergence between marginal distributions of the terminal parameter is upper bounded by the KL between the distributions of the full trajectories.

Assuming $Q_0 = P_0$, we may decompose $\text{KL}(Q \| P)$ across iterations, obtaining

$$\text{KL}(Q \| P) = \underset{W \sim Q}{\mathbb{E}} \left[ \log \frac{\text{d}Q}{\text{d}P}(W) \right] = \underset{W \sim Q}{\mathbb{E}} \left[ \sum_{t=1}^{T} \log \frac{\text{d}Q_{t|}}{\text{d}P_{t|}}(W) \right] = \sum_{t=1}^{T} \mathbb{E}_{Q_{0:(t-1)}}[\text{KL}(Q_{t|} \| P_{t|})]. \quad (13)$$

□

# C  Mutual Information Bound for Subgaussian Losses

**Theorem C.1** (Xu and Raginsky's Theorem 1). *Suppose that $\ell(w,Z)$ is $\sigma$-subgaussian when $Z \sim \mu$, for all $w \in \mathscr{W}$, Then*

$$\left| \mathbb{E} \left[ R_{\mathscr{D}}(W) - \hat{R}_S(W) \right] \right| \leq \sqrt{\frac{2\sigma^2}{n} I(S;W)}$$

A proof of this result is found in [35]. However, one may use the arguments therein to establish the further conclusion that $\ell(W,Z)$ or $R_S(W)$ is also subgaussian, which is not generally true. In this section we briefly describe the flaw in that logic and provide a clarification of their proof under the same assumptions. [29] give a proof for discrete parameter spaces, which does not contain this flaw. While it is straightforward to cast their proof into measure-theoretic language, we give the details for completeness.

The discussion in [35] preceding the theorem asserts that if $f : \mathscr{W} \times \mathscr{S}$ is such that $f(w,S)$ is $\sigma$ subgaussian for all $w \in \mathscr{W}$ and if $W \perp\!\!\!\perp S$ then $f(W,S)$ is $\sigma$-subgaussian. A simple counter example is given by $\mathscr{W} = \mathscr{S} = \mathbb{R}$, with $f(w,s) = w + s$, and $(W,S) \sim \text{Cauchy} \times N(0,1)$. In this case $f(w,s)$ is clearly 1-subgaussian for each $w \in \mathscr{W}$, while $f(W,S)$ does not even have bounded absolute first moment, let alone a moment generating function defined in any open ball about 0.

The main issue in the argument establishing subgaussianity of $f(W,S)$ is failing to properly use a version of the conditional variance formula (modified to apply for moment generating functions as opposed to variances). The intuition of the conditional variance formula is useful in reconciling the final result with our counterexample, but is not sufficient for a general proof as the subgaussian parameter is not generally a standard deviation. The conditional variance formula asserts that

$$\text{Var}(f(W,S)) = \mathbb{E} \left[ \text{Var}^W f(W,S) \right] + \text{Var} \left( \mathbb{E}^W f(W,S) \right).$$

The argument by which one would conclude that $f(W,S)$ is subgaussian only acknowledges the first term, thus assuming that the second term is 0 (which would only hold when $\mathbb{E}^W f(W,S)$ is a.s. constant in $W$).

More precisely, since we are working with subgaussian parameters instead of true standard deviations:

$$\begin{aligned}
&\log \mathbb{E} \exp(t(f(W,S) - \mathbb{E}f(W,S))) \\
&= \log \mathbb{E} \left[ \exp(t(\mathbb{E}^W f(W,S) - \mathbb{E}f(W,S)))\mathbb{E}^W \exp(t(f(W,S) - \mathbb{E}^W f(W,S))) \right] \\
&\leq \log \exp(t^2\sigma^2/2)\mathbb{E}[\exp(t(\mathbb{E}^W f(W,S) - \mathbb{E}f(W,S)))] \\
&= t^2\sigma^2/2 + \log \mathbb{E}[\exp(t(\mathbb{E}^W f(W,S) - \mathbb{E}f(W,S)))]
\end{aligned}$$

The RHS is $\geq t^2\sigma^2/2$ with equality if and only if $(\mathbb{E}^W f(W,S) - \mathbb{E}f(W,S))$ is constant (by Jensen' inequality). The first inequality is an equality when $f(w,S)$ is normal with variance $\sigma^2$ for all $w \in \mathcal{W}$.

Ergo, the assertion that $f(W,S)$ is $\sigma$-subgaussian holds exactly when $(\mathbb{E}_S f(W,S) - \mathbb{E}f(W,S))$ is constant. This situation is not generally of interest in learning theory; this amounts to saying that all parameter vectors lead to the same expected generalization error, and hence there is no purpose to learning from the data!

The final result is, of course, still valid and may be proven directly via the Donsker–Varadhan variational formula.

*Proof.* As in [35] we will leverage the fact that for each $w \in \mathcal{W}$, $f(w,S) = \frac{1}{n}\sum_{i=1}^{n}\ell(w,Z_i)$ is $\tau = \sigma/\sqrt{n}$ subgaussian, *however these variable may have different means for each value of w.* Let $\check{f}(w,s) = f(w,s) - \mathbb{E}f(w,S)$.

By Donsker–Varadhan and the fact that $\mathbb{E}^W \check{f}(\bar{W},\bar{S}) = 0$ a.s.,

$$I(W;S) \geq \mathbb{E}\lambda\check{f}(W,S) - \log\mathbb{E}\exp(\lambda\check{f}(\bar{W},\bar{S}))$$
$$\geq \mathbb{E}\lambda\check{f}(W,S) - \log\mathbb{E}\mathbb{E}^W\exp(\lambda\check{f}(\bar{W},\bar{S}))$$
$$\geq \lambda\mathbb{E}\check{f}(W,S) - \log\mathbb{E}\exp(\lambda^2\tau^2/2)$$
$$\geq \lambda\mathbb{E}\check{f}(W,S) - \lambda^2\tau^2/2.$$

Optimizing over $\lambda$ now yields the desired result, because

$$|\mathbb{E}\check{f}(W,S)| = |\mathbb{E}\left[f(W,S) - \mathbb{E}^W f(W,\bar{S})\right]| = |\mathbb{E}\left[R_{\mathscr{D}}(W) - \hat{R}_S(W)\right]|.$$

$\square$

# D  Properties of the Hypergeometric Distribution and of Finite Population Variances

In this section, we enumerate a number of well-known results, and also derive some particular ones for our application.

## D.1  Properties of the Hypergeometric Distribution

Let $n,m,b \in \mathbb{N}$, $m,b \leq n$. Write $B \sim \text{HG}(n,m,b)$ when

$$\mathbb{P}(B = j) = \frac{\binom{m}{j}\binom{n-m}{b-j}}{\binom{n}{b}}, \quad j \in \{0 \vee b+m-n,\ldots,n\wedge m\}.$$

It follows that

$$\mathbb{E}(B) = b\frac{m}{n} \qquad \text{Var}(B) \qquad = b\frac{m}{n}\frac{n-m}{n}\frac{n-b}{n-1} \leq b\frac{m(n-m)}{n^2}$$

## D.2  Finite Population Statistics with Disjoint Samples

In this section we compute the covariance of the sample means for each population, and provide a formula for the variance of a linear combination of the two estimators.

**Lemma D.1** (Variance for disjoint finite population statistics). *Suppose that there is a finite population of size, N, $S = (y_1,...,y_N)$. Consider two disjoint subsets of sizes $n_1$ and $n_2$ are chosen uniformly at random from S. Let $\bar{Y}_i$ be the sample mean on the ith sample. Let $\Sigma$ be the population variance matrix. Then*

$$Var\begin{pmatrix}\bar{Y}_1 \\ \bar{Y}_2\end{pmatrix} = \frac{1}{N-1}\begin{bmatrix}(N-n_1)/n_1 & -1 \\ -1 & (N-n_2)/n_2\end{bmatrix} \otimes \Sigma$$

$$Var(a\bar{Y}_1 - b\bar{Y}_2) = \frac{1}{(N-1)}\left(-(a-b)^2 + N(a^2/n_1 + b^2/n_2)\right)\Sigma$$

*Proof.* Let $\zeta_i$ be an indicator for whether $y_i$ appears in the first sample, and let $W_i$ be an indicator for whether $y_i$ appears in the second sample.

Let $\mu = \frac{1}{N}\sum_{i=1}^{N} y_i$ and let $\Sigma = \frac{1}{N}\sum_{i=1}^{N}(y_i - \mu)(y_i - \mu)'$

Then for any $i \neq j$:

$$\zeta_i \sim \mathrm{Ber}(n_1/N) \qquad\qquad W_i \sim \mathrm{Ber}(n_2/N)$$

$$\mathrm{Var}(\zeta_i) = \frac{n_1(N-n_1)}{N^2} \qquad\qquad \mathrm{Var}(W_i) = \frac{n_2(N-n_2)}{N^2}$$

$$
\begin{aligned}
\mathrm{Cov}(\zeta_i,\zeta_j) &= \mathbb{E}[\zeta_i\zeta_j] - \frac{n_1^2}{N^2} \\
&= \mathbb{P}[\zeta_i = \zeta_j = 1] - \frac{n_1^2}{N^2} \\
&= \frac{n_1(n_1-1)}{N(N-1)} - \frac{n_1^2}{N^2} \\
&= -\frac{n_1}{N}\left(1 - \frac{n_1}{N}\right)\frac{1}{N-1}
\end{aligned}
\qquad
\begin{aligned}
\mathrm{Cov}(W_i,W_j) &= \mathbb{E}[W_iW_j] - \frac{n_2^2}{N^2} \\
&= \mathbb{PP}[W_i = W_j = 1] - \frac{n_2^2}{N^2} \\
&= \frac{n_2(n_2-1)}{N(N-1)} - \frac{n_2^2}{N^2} \\
&= -\frac{n_2}{N}\left(1 - \frac{n_2}{N}\right)\frac{1}{N-1}
\end{aligned}
$$

$$
\begin{aligned}
\mathrm{Cov}(\zeta_i,W_i) &= \mathbb{E}[\zeta_iW_i] - \frac{n_1n_2}{N^2} \\
&= \mathbb{P}[\zeta_i = W_j = 1] - \frac{n_1n_2}{N^2} \\
&= 0 - \frac{n_1n_2}{N^2} \\
&= -\frac{n_1n_2}{N^2}
\end{aligned}
\qquad
\begin{aligned}
\mathrm{Cov}(\zeta_i,W_j) &= \mathbb{E}[\zeta_iW_j] - \frac{n_1n_2}{N^2} \\
&= \mathbb{PP}[\zeta_i = W_j = 1] - \frac{n_1n_2}{N^2} \\
&= \frac{n_1n_2}{N(N-1)} - \frac{n_1n_2}{N^2} \\
&= \frac{n_1n_2}{N^2(N-1)}.
\end{aligned}
$$

$$(\bar{Y}_1, \bar{Y}_2) = \sum_{i=1}^{N} y_i(\zeta_i/n_1, W_i/n_2)$$

$$
\begin{aligned}
\mathrm{Var}(\bar{Y}_1) &= \mathrm{Var}\left(\sum_{i=1}^{N}\frac{y_i}{n_1}\zeta_i\right) \\
&= \frac{1}{n_1^2}\left(\sum_{i=1}^{N} y_iy_i'\frac{n_1(N-n_1)}{N^2} - \sum_{i\neq j} y_iy_j'\frac{n_1(N-n_1)}{N^2(N-1)}\right) \\
&= \frac{(N-n_1)}{n_1N^2}\left(\sum_{i=1}^{N} y_iy_i' - \sum_{i\neq j} y_iy_j'\frac{1}{N-1}\right) \\
&= \frac{(N-n_1)}{n_1(N-1)N}\sum_{i=1}^{N}(y_i - \mu)(y_i - \mu)' \\
&= \frac{(N-n_1)}{n_1(N-1)}\Sigma
\end{aligned}
$$

Similarly

$$\mathrm{Var}(\bar{Y}_2) = \frac{(N-n_2)}{n_2(N-1)}\Sigma$$

Now, for the less well known part:

$$\mathrm{Cov}(\bar{Y}_1, \bar{Y}_2) = \mathrm{Cov}\left(\sum_{i=1}^{N} \frac{y_i}{n_1}\zeta_i, \sum_{i=1}^{N} \frac{y_i}{n_2}W_i\right)$$

$$= \sum_{i=1}^{N} \frac{y_i y_i'}{n_1 n_2}\mathrm{Cov}(\zeta_i, W_i) + \sum_{i \neq j} \frac{y_i y_j'}{n_1 n_2}\mathrm{Cov}(\zeta_i, W_j)$$

$$= -\sum_{i=1}^{N} \frac{y_i y_i'}{n_1 n_2}\frac{n_1 n_2}{N^2} + \sum_{i \neq j} \frac{y_i y_j'}{n_1 n_2}\frac{n_1 n_2}{N^2(N-1)}$$

$$= -\frac{1}{N^2}\left(\sum_{i=1}^{N} y_i y_i' - \sum_{i \neq j} y_i y_j'\frac{1}{N-1}\right)$$

$$= -\frac{1}{N-1}\Sigma$$

Hence

$$\mathrm{Var}\begin{pmatrix}\bar{Y}_1\\\bar{Y}_2\end{pmatrix} = \frac{1}{N-1}\begin{bmatrix}(N-n_1)/n_1 & -1\\ -1 & (N-n_2)/n_2\end{bmatrix} \otimes \Sigma$$

For our application we need $\mathrm{Var}(a\bar{Y}_1 - b\bar{Y}_2)$:

$$\mathrm{Var}(a\bar{Y}_1 - b\bar{Y}_2) = a^2\frac{(N-n_1)}{n_1(N-1)}\Sigma + b^2\frac{(N-n_2)}{n_2(N-1)}\Sigma + 2ab\frac{1}{N-1}\Sigma$$

$$= \frac{1}{(N-1)}\left(a^2\frac{N-n_1}{n_1} + 2ab + b^2\frac{N-n_2}{n_2}\right)\Sigma$$

$$= \frac{1}{(N-1)}\left(-(a-b)^2 + N(a^2/n_1 + b^2/n_2)\right)\Sigma$$

$\square$

**Lemma D.2** (Bounding $\mathbb{E}\mathbb{E}^{S_J,J,U}\|\xi_t\|_2^2$ for SGLD)**.** *In the setting of Section* 3.1

$$\mathbb{E}\mathbb{E}^{S_J,J,U}\|\xi_t\|_2^2 = \frac{n(n-m)}{(n-1)^2 b_t}\left(1 + \frac{b_t}{n}\frac{n-m-1}{m}\right)\mathbb{E}[\hat{\Sigma}_t(S)]$$

*Proof.* Applying the conditional variance formula gives:

$$\mathbb{E}\mathbb{E}^{S_J,J,U}\|\xi_t\|_2^2 = \mathbb{E}\mathrm{Var}^{S,W_t}(\mathbb{E}^{b_t,W_t,S}[\xi_t]) + \mathbb{E}\mathbb{E}^{S,W_t}[\mathrm{Var}^{b_t,W_t,S}(\xi_t)]$$

$$= 0 + \mathbb{E}\mathbb{E}^{S,W_t}\left[\mathrm{Var}^{b_t,W_t,S}\left(\frac{b_t^c}{b_t}\nabla\tilde{R}_{S_t^c}(W_t) - \frac{b_t^c}{b_t}\nabla\tilde{R}_{S_J}(W_t)\right)\right]$$

$$= \mathbb{E}\mathbb{E}^{S,W_t}\left[\frac{(b_t^c)^2}{b_t^2}\mathrm{Var}^{b_t,W_t,S}\left(\nabla\tilde{R}_{S_t^c}(W_t) - \nabla\tilde{R}_{S_J}(W_t)\right)\right]$$

Applying Lemma D.1 further yields

$$\mathbb{E}^{S,W_t}\left[\frac{(b_t^c)^2}{b_t^2}\mathrm{Var}^{b_t,W_t}\left(\nabla\tilde{R}_{S_t^c}(W_t)-\nabla\tilde{R}_{S_J}(W_t)\right)\right]$$

$$=\mathbb{E}^{S,W_t}\left[\frac{(b_t^c)^2}{b_t^2}\frac{1}{(n-1)}\left(\frac{n}{b_t^c}+\frac{n}{m}\right)\hat{\Sigma}_t(S)\right]$$

$$=\frac{n}{(n-1)b_t^2}\mathbb{E}^{S,W_t}\left[b_t^c+(b_t^c)^2\frac{1}{m}\right]\mathbb{E}^{S,W_t}[\hat{\Sigma}_t(S)]$$

$$=\frac{n}{(n-1)b_t^2}\left(b_t\frac{n-m}{n}+\left(\frac{(n-m)^2}{n^2}b_t^2+b_t\frac{m}{n}\frac{n-m}{n}\frac{n-b_t}{n-1}\right)\frac{1}{m}\right)\mathbb{E}^{S,W_t}[\hat{\Sigma}_t(S)]$$

$$=\frac{n}{(n-1)b_t^2}\left(b_t\frac{n-m}{n-1}+b_t^2\frac{(n-m)(n-m-1)}{n(n-1)m}\right)\mathbb{E}^{S,W_t}[\hat{\Sigma}_t(S)]$$

$$=\frac{n}{(n-1)b_t^2}\left(b_t\frac{n-m}{n-1}+b_t^2\frac{(n-m)(n-m-1)}{n(n-1)m}\right)\mathbb{E}^{S,W_t}[\hat{\Sigma}_t(S)]$$

$$=\frac{n(n-m)}{(n-1)^2b_t}\left(1+\frac{b_t}{n}\frac{n-m-1}{m}\right)\mathbb{E}^{S,W_t}[\hat{\Sigma}_t(S)]$$

$\square$

# E    Asymptotic Results

## E.1    Langevin Dynamics

In this section we continue from the end of Section 3.2.1. Under the assumption that $\tilde{\ell}$ is $L$-Lispchitz (the same assumption as in [6]) we have the following results which portray the asymptotic behavior of the expected generalization error of the Langevin diffusion algorithm for $\ell$ being the 0-1 loss (which is $1/2$-subgaussian):

$$\mathbb{E}_{W_T\sim Q_T}(R_{\mathscr{D}}(W_T)-R_S(W_T))\leq\frac{L}{2(n-1)}\sqrt{\sum_{t=1}^{T}\beta_t\eta_t}$$

### E.1.1    Geometrically Decaying Learning Rate

Under an assumption of $L$-Lipschitz loss and geometrically decaying learning rate and a temperature that ramps up to a polynomial in $n$ ($\eta_t=\eta_0\rho^t$ for $0<\rho<1$ and that $\beta_t=\beta_0(n-1)^\theta(1-\nu^t)$ for some $0<\theta<1$) then we have the following bound:

$$\sup_{T\geq 0}\left[\mathbb{E}_{W_T\sim Q_T}(R_{\mathscr{D}}(W_T)-R_S(W_T))\right]\leq\frac{L}{2(n-1)^{1-\theta}}\sqrt{\beta_0\eta_0\frac{\rho(1-\nu)}{(1-\rho)(1-\rho\nu)}}$$

### E.1.2    Polynomial Decaying Learning Rate

Under an assumption of $L$-Lipschitz loss and polynomial decaying learning rate and temperature that is polynomial in $n$ ($\eta_t=\eta_0t^{-\alpha}$ for $\alpha>0$ and that $\beta_t=\beta_0(n-1)^p$ for some $0<p<1$) then we have the following bound:

$$\left[\mathbb{E}_{W_T\sim Q_T}(R_{\mathscr{D}}(W_T)-R_S(W_T))\right]\leq\begin{cases}\frac{L}{2(n-1)^{1-p}}\sqrt{1+\frac{1}{\alpha-1}T^{1-\alpha}}&\alpha<1\\\frac{L}{2(n-1)^{1-p}}\sqrt{1+\log(T)}&\alpha=1\\\frac{L\alpha}{2(n-1)^{1-p}(\alpha-1)}&\alpha>1\end{cases}$$

# F    Comparing Theorems 2.3 to 2.5 when $m=n-1$

Let $V\sim P(S_J,U)$, $W\sim Q(S,U)$, and $\tilde{W}\sim Q(S,U)$ independently of $W$. In the case of $[a_1,a_2]$-bounded loss, $\left(R_{\mathscr{D}}(V)-\hat{R}_{S\backslash S}(V)\right)$ is $(a_2-a_1)/2$-subgaussian, so that Theorem 2.5 yields:

$$\mathbb{E}\left[R_{\mathscr{D}}(W)-\hat{R}_S(\tilde{W})\right]\leq\mathbb{E}\sqrt{(a_2-a_1)^2\,\mathrm{KL}(Q(S,U)\,\|\,P(S_J,U))\,/2}.$$

Using KL divergence based upper bounds for mutual information (Proposition B.1), Theorem 2.4 gives us

$$\mathbb{E}\left[R_{\mathscr{D}}(W) - \hat{R}_S(W)\right] \leq \mathbb{E}\sqrt{(a_2 - a_1)^2 \mathbb{E}^{S_J,U}\left[\mathrm{KL}(Q(S,U) \,\|\, P(S_J,U))\right]/2},$$

while Theorem 2.3 yields:

$$\mathbb{E}\left[R_{\mathscr{D}}(W) - \hat{R}_S(W)\right] \leq \sqrt{(a_2 - a_1)^2 \mathbb{E}\left[\mathrm{KL}(Q(S,U) \,\|\, P(S_J,U))\right]/2}$$

for $m = n - 1$, the bounds are ranked as $2.3 \geq 2.4 \geq 2.5$ (by Jensen's inequality for each conditional expectation being passed into $\sqrt{\cdot}$). When $\mathrm{KL}(Q(S) \,\|\, P(S_J))$ has a large variance then the difference can be quite material.

# G   An analytically tractable example

We present a simple analytic example, where our upper bound is a clear improvement over existing work when similar simplifications are performed. Let $S = \{z_1, \ldots, z_n\} \sim \mathscr{D}^n$ be a sample from the distribution $\mathscr{D}$ on $\mathbb{R}$. We wish to estimate the mean of $\mathscr{D}$, $\mu$. We will use the loss function $\ell(z, w) = \tilde{\ell}(z, w) = (z - w)^2$ where $w \in \mathscr{W} = \mathbb{R}$. The distribution $\mathscr{D}$, is assumed to satisfy the sub-Gaussianity assumption in Theorems 2.3 and 2.5 for this loss. Upon specializing the SGLD update rule (7) to this setting:

$$W_{t+1} = W_t - \eta_t \frac{d}{dW_t}\tilde{R}_S(W_t) + \sqrt{\frac{2\eta_t}{\beta}}\varepsilon_t = \left(1 - \frac{2\eta_t}{n}\right)W_t + \frac{2\eta_t}{n}\sum_{i=1}^{n} z_i + \sqrt{\frac{2\eta_t}{\beta}}\varepsilon_t. \quad (14)$$

We will apply the data-dependent generalization bound in Theorem 2.5 with $m = \#S_J = n - 1$ and set $\{i^\star\} = J$. Since we are working with LD, we set the random variable $U$ to a constant (trivial random variable). It follows that:

$$\mathrm{KL}(Q_{t+1|}(S) \,\|\, P_{t+1|}(S_J)) = \frac{(\mu_{t+1} - \mu'_{t+1})^2}{4\eta_t/\beta} = \frac{\beta}{n^2}z_{i^\star}^2\eta_t. \quad (15)$$

Thus the expected generalization error is upper bounded by:

$$\mathbb{E}\sqrt{2\sigma^2 KL(Q_T(S)\|P_T(S_J))} \leq \mathbb{E}\sqrt{2\sigma^2 \frac{\beta}{n^2}z_{i^\star}^2 \sum_{t=0}^{T-1}\eta_t} = \mathbb{E}[|z_i|]\left(\sqrt{2\sigma^2 \frac{\beta}{n^2}\sum_{t=0}^{T-1}\eta_t}\right). \quad (16)$$

When one applies the results in [24, 35], the upper bounded on the generalization error can be shown to be:

$$\sqrt{\frac{2\sigma^2}{n}I(W_T;S)} \leq \sqrt{\frac{2\sigma^2}{n}\sum_{t=0}^{T-1}I(\bar{W}_{t+1};S|W_1^t)} \leq \sqrt{2\sigma^2\frac{\beta}{n^2}E[z_i^2]\sum_{t=0}^{T-1}\eta_t} \quad (17)$$

Comparing with (16) we see that this bound can be is larger since $E[|z_i|] \leq \sqrt{E[z_i^2]}$ from Jensen's inequality. The discrepancy can be made arbitrarily large based on the choice of $\mathscr{D}$.

# H   Experiment Details

The first architecture and dataset we consider is a three-layer multilayer perceptron (MLP), with 600 hidden units per hidden layer and rectified linear unit (ReLU) activation functions, trained on MNIST [19]. In Fig. 1a, we compare the bound for two amounts of held out data, $n - m = \#S_J^c$. We see that the empirical performance reflects our analytical results that the bound is tighter for large $m$. As can be inferred from Eq. (4), the difference between $\|\xi_t\|^2$ and $\|\nabla\tilde{R}_t\|^2$ increases with $m$.

The remainder of our experiments consider convolutional neural networks (CNNs). For MNIST and Fashion-MNIST, we use a standard network configuration with two convolutional layers (with 32 and 64 filters of size $5 \times 5$, respectively, followed by $2 \times 2$ max pooling after each convolutional layer), followed by two fully connected layers (1024 nodes each) with ReLU activations.

Our final experiment uses the CIFAR-10 dataset. The CNN architecture has two convolutional layers and three fully connected layers. Both convolutional layers use 64 filters of size $5 \times 5$. After each convolutional layer there is a $2 \times 2$ max pooling layer. Then, we have three fully connected layers with the number of neurons 384, 192, and 10 respectively.

### H.1 Evaluation of the generalization bound

We estimate our generalization error bound and that of [22] using nested Monte Carlo simulations. We use the results of Theorem 3.1, specifically Eq. (6). In order to evaluate this bound we perform two Monte Carlo estimate: one for $\mathbb{E}^{S,J,U}\|\xi_t\|_2^2$, and then for the full expectation (outside of the $\sqrt{\cdot}$). For our bound, for each hyperparameter combination, we have used 10 simulations for the outer expectation, each using 10 simulations to estimate the inner expectation. For the generalization bound in Mou et al. [22] we have used their 100 simulations to evaluate the bound given by their Theorem 10.

### H.2 Learning Rate and Inverse Temperatures for Figs. 1b and 1c

In Fig. 1b, we use

$$\beta_t^{(\text{high})} = 100 \times \max\{\exp\left(\frac{t}{100}\right), 55000\} \tag{18}$$

$$\beta_t^{(\text{low})} = 100 \times \max\{\exp\left(\frac{t}{100}\right), 5000\} \tag{19}$$

where $t$ denotes the iteration number.

All other parameters are the same and are outlined in Table 2.

For the "high' inverse temperature schedule, at Iteration 5 the training error is 3.21% and the generalization error is 0.9%, while for the "low' inverse temperature schedule, at epoch 5 the training error is 5.18% and the generalization error is 0.16%.

In Fig. 1c we consider $\eta_t^{(\text{small})} = 8 \times 10^{-4} \times 0.96^{\left(\frac{t}{2000}\right)}$ and $\eta_t^{(\text{large})} = 2 \times 10^{-3} \times 0.96^{\left(\frac{t}{2000}\right)}$, and the rest of the parameters are the same and are outlined in Table 2. For the "small" learning rate, the training error and the test-set generalization error at Epoch 6 for the small learning rate scenario are 7.62% and 1.1% , respectively,; while for the "large" learning rate the training error and the test-set generalization error at Epoch 6 are 6.3% and 1.0%, respectively.

### H.3 Hyperparameters of our experiments

In Tables 1 to 4, we provide the hyperparameter and training details of the experiments that were presented in Section 4.

| Parameter | Values |
|---|---|
| Dataset | MNIST |
| Architecture | MLP with 3 hidden layers |
| Batch size | 100 |
| Learning rate | `learning rate=`$8 \times 10^{-3}$`,decay steps=600, decay rate=0.95` |
| Beta schedule | $\min\{10 \times \exp\left(\text{iter}/400\right), 2000\}$ |
| Number of epochs | 15 |
| Average Final training error | 1.40% |
| Average Final test error | 4.12% |
| # training examples | 55000 |
| Number of runs | 50 |

Table 1: Details of Experiments reported in Fig. 1a for MNIST with MLP

## I High Probability PAC-Bayes Bounds

We can leverage the methods used to provide bounds on the expected generalization error above to also derive high probability bounds for the generalization error. We will give an example of this here for completeness, though more work can be done to select a tighter bound from more recent literature and to tune the parameters available to optimize the bound further. For example, in our setting we could optimally tune the level of data dependence for the bound to be tightened. We will

| Parameter | Values |
|---|---|
| Dataset | MNIST |
| Architecture | CNN with 2 conv. layers |
| Batch size | 100 |
| Learning rate | `learning rate=4 × 10⁻³,decay steps=2000, decay rate=0.96` |
| Beta schedule | $\min\{10 \times \exp(\text{iter}/100), 55000\}$ |
| Number of epochs | 15 |
| Average Final training error | 1.81% |
| Average Final test error | 2.03% |
| # training examples | 55000 |
| Number of runs | 50 |

Table 2: Details of Experiments reported in Figs. 1b to 1d for MNIST with CNN

| Parameter | Values |
|---|---|
| Dataset | Fashion-MNIST |
| Architecture | CNN with 2 conv. layers |
| Batch size | 100 |
| Learning rate | `learning rate=4 × 10⁻³,decay steps=3500, decay rate=0.93` |
| Beta schedule | $\min\{10 \times \exp(\text{iter}/100), 55000\}$ |
| Number of epochs | 25 |
| Average Final training error | 8.3% |
| Average Final test error | 10.83% |
| # training examples | 60000 |
| Number of runs | 20 |

Table 3: Details of Experiments reported in Fig. 1e for Fashion-MNIST

make use of Shalev-Shwartz and Ben-David [30] (theorem 31.1 therein), which we state here under the notation and definitions of our work, and in the context of Section 3.1.

**Proposition I.1** ([30] Theorem 31.1). *Suppose that the loss function is $[0,1]$-bounded. Let $P$ be any prior distribution. With probability at least $(1-\delta)$ (over the choice of $S \sim \mathscr{D}^n$) for any posterior distribution $Q$ (even those depending on $S$) with $W \sim Q$,*

$$\mathbb{E}^S\left[R_{\mathscr{D}}(W_T) - \hat{R}_S(W_T)\right] \leq \sqrt{\frac{\mathrm{KL}(Q\,\|\,P) + \log(n/\delta)}{2(n-1)}}$$

In our setting $P$ will be allowed to depend on $m$ data points chosen uniformly at random, while $Q$ will depend on the full dataset, so we can apply this result conditional on the subset upon which $P$ depends. Therefore, for any $S_J \in \mathscr{Z}^m$ and any $U \in \mathscr{U}$ and for any kernel $P : \mathscr{Z}^n \times \mathscr{U} \to \mathscr{M}_1(\mathscr{W}^T)$ be any prior distribution which depends on $S_J$, with probability at least $(1-\delta)$ (over the choice of $S_J^c \sim \mathscr{D}^{n-m}$) for any posterior distribution $Q$ (even those depending on $S_J^c$) with $W \sim Q$,

$$\mathbb{E}^S\left[R_{\mathscr{D}}(W_T) - \hat{R}_{S_J^c}(W_T)\right] \leq \sqrt{\frac{\mathrm{KL}(Q(S,U)\,\|\,P(S_J,U)) + \log((n-m)/\delta)}{2(n-m-1)}}$$

$$\leq \sqrt{\frac{\sum_{t=1}^{T} \frac{\beta_t \eta_t}{8} \mathbb{E}^{S,J,U}\|\xi_t\|_2^2 + \log((n-m)/\delta)}{2(n-m-1)}}$$

In the case of Langevin dynamics when using worst case, Lipschitz constant base upper bounds, this gives

$$\mathbb{E}^S\left[R_{\mathscr{D}}(W_T) - \hat{R}_{S_J^c}(W_T)\right] \leq \sqrt{\frac{\frac{L^2}{(n-1)(m-1)}\sum_{t=1}^{T}\frac{\beta_t \eta_t}{8} + \log((n-m)/\delta)}{2(n-m-1)}}$$

which provides a less trivial tradeoff between $m$ and $n-m$ compared to the expected generalization error bound. One could further take expectations over $U$ and/or $J$ to get high probability bounds for the generalization error based on the full empirical loss.

| Parameter | Values |
|---|---|
| Dataset | CIFAR-10 |
| Architecture | CNN with 2 conv. layers |
| Batch size | 200 |
| Learning rate | `learning rate=`$5 \times 10^{-3}$`,decay steps=2000, decay rate=0.95` |
| Beta schedule | $\min\{10 \times \exp\left(\text{iter}/100\right), 55000\}$ |
| Number of epochs | 50 |
| Average Final training error | 6.9% |
| Average Final test error | 29.9% |
| $|S_J|$ | `len(training_set)-1` |
| # training examples | 50000 |
| Number of runs | 30 |

Table 4: Details of Experiments reported in Fig. 1f for CIFAR-10

We intend to investigate such bounds further in future work, and this section serves merely to illustrate the possibility and nature of such high-probability bounds based on data-dependent estimates of mutual information and data-dependent PAC-Bayes priors. We acknowledge that these are not the tightest such bounds possible.