[Reviews · NeurIPS 2019]

Reviewer 1



This work leverages concepts from PAC-Bayes style generalization bounds to the estimation of local mutual information in neural networks and its application to bounding generalization (similar to Xu and Raginsky). Unfortunately there are numerous spelling and grammar mistakes, and more problematically, critical notation in the main theorems are not defined anywhere. Additionally, important quantities are glossed over and not discussed thoroughly enough. As a result, the theoretical sections are quite difficult to follow. It is not clear to me how the information bounds are used, it seems that instead of these bounds the authors end up focusing on KL-based bounds which are more reminiscent of PAC-Bayes. For the experimental results, the improvement over non-data-dependent bounds is to be expected. The figure should be relabeled, "Numerical Results" is not informative, and the y axis should be labeled. Edit: Thanks for answering my questions. The authors seem willing to put in the effort for properly revising the paper, so I tend towards acceptance.

Reviewer 2



This looks like a good set of theoretical contributions. I found the paper a fair bit harder to read than the previous papers on information-theoretic generalization, and I have very limited knowledge of the PAC-Bayes literature, so overall my confidence is at most “medium”. Although I would typically focus on theoretical contributions, in this case the experimental validation seems to play quite an important role, since without this it is unclear how much improvement there is (if any) over existing bounds. This leads me to the main issue that comes to mind – the notion of comparing the “key quantities” in the generalization bounds is considerably less convincing compared to comparing the bounds themselves. Why can’t the latter be done? (If it is somehow impossible, this ought to be explained in the paper) It should also be established that your generalization bounds are non-trivial, since a comparison of the form “trivial” vs. “a few orders of magnitude above trivial” would be somewhat less valuable. Aside from that, my main suggestions would be to try to make the paper (including the appendix) more accessible to less experienced readers. Some specific suggestions: - Define all notation unless it is very standard (e.g., \calx^{[T]}, \calM_1(\calW), #S_{J_t}), and be careful of undefined symbols even if their meaning is clear (e.g., W, and similarly \bar{W} and \bar{S} on p13) - Include citations/references even for fairly standard formulas (e.g., sample variances in finite populations on p5, the chain rule for KL divergence in (53), the Donsker-Varadhan formulation many times in the appendix – which should be added as a display equation, etc.) - For better visibility, and to make all the relevant assumptions clear, perhaps state (16) and (26) as “Corollary 1” and “Corollary 2”? - Do not skip non-trivial details in the appendix. For example, the “reduces to” claim leading to (34) may not be immediate for typical readers. - Check for inconsistencies in the main body and appendix. For example, Proposition 2.4 is written in terms of P_T(omega) but then P_T has no argument in the appendix, which is confusing. Other comments: - The reference list seems a bit too short given the amount of related work (on information-theoretic generalization, PAC-Bayes, Langevin algorithms, etc.). - It is not acceptable to define key quantities/notations (e.g., loss function, mu_X notation) in the appendix and then go ahead and use them in the main body. - Before (32), I don’t see how the “RHS is constant in J”. What if W depends more strongly on certain indices than others? - It is nice to have the flaw outlined in Appendix C formally documented. I think it’s worth adding (if I’m not mistaken) that, as an alternative to your fix on p13, Russo and Zou’s approach also leads to the same result without any errors. - Has [11] been published? If so, please modify. If not, consider switching to the AISTATS reference. Minor comments: - The second-last paragraph before Section 1.1 is hard to follow - Section 1.1: Add “We” at the start of each dot point to form a proper sentence - Section 3.1.1: Too many “and let” in a row - Typo: “These bound elucidates” - p12: Mention Radon-Nikodym derivative for less experienced readers === POST-REBUTTAL: Thank you for the responses. While I am a bit reluctant as to whether the required changes are minor enough for acceptance with no re-review, I have upgraded to 7. If accepted, the authors should very carefully address the reviewer comments, including much more strongly highlighting (i.e., not buried in an appendix) the numerical and analytical examples where the *generalization bound* is improved and non-vacuous.

Reviewer 3



The paper studies the problem of data-dependent generalization bounds. Novel abstract results are presented, which extend the existing mutual information and PAC-Bayes bounds, which scale with the mutual information or KL divergence related to a random uniform subsample of the original dataset. Based on this framework, an improved generalization bound for the Langevin algorithm is developed, which achieves $O(1/n)$ rate and depends only on the trace of covariance of stochastic gradients. The idea of proof is simple but very interesting. Mutual information method and PAC-Bayes bounds are both based on Donsker-Varadhan variational formula. Plugging in one of the measures to be based on a random subsample brings the ideas of leave-one-out analysis to this literature. Such type of extension is nontrivial and can potentially bring about better understanding of data-dependent generalization bounds. The examples of Langevin dynamics and SGLD also made solid progress in this direction. That being said, the current presentation impedes readers' understanding of the techniques, and the writing needs a lot of improvement. For example, in Theorem 2.1, it is not clear what the function $\psi_{R_D(\tilde{W}) - \hat{R}_{S_J^C} (\tilde{W})} (\lambda)$ means. The results about Langevin dynamics and SGLD should be formally stated as corollaries or theorems. Besides, the authors should also include more intuition about the meaning of subsample J and auxiliary random variable X.

[Author Response · NeurIPS 2019]

1. All reviewers. We acknowledge the paper uses some unusual notation and, in parts, assumes background in information theory. We now give all standard definitions. We have corrected typos and grammatical errors.

2. Reviewer #1 asks how the mutual information bounds are used and why we seem to focus on KL bounds, which resemble PAC-Bayes bounds. Our SGLD bounds are built using Theorem 2.2 (mutual inf. based) and Theorem 2.3 (KL based). Note that the mutual information in Thm. 2.2 involves unknown distributions and cannot be computed directly. We get explicit bounds on mutual informations via the known identity $I(X;Y) = \inf_P \mathbb{E}[\mathrm{KL}(Q(X)\|P)]$, where $Q(X)$ is the conditional law of $Y$ given $X$, and $P$ ranges over all distributions on the same space as $Y$. (The $P$s are called "priors" in the PAC-Bayes literature.) Eqs. (8),(9) exploit this identity. We now highlight this identity and explain its role clearly. One of our key contributions is inventing data-dependent priors (i.e., the prior depends on $S_J$) that yield explicit bounds that are much more adapted to the true unknown distribution, and thus yield much tighter bounds. These same data-dependent priors appear in Thm. 2.3.

3. Reviewer #2 points out some missing related work. First, we acknowledge and thank the reviewer for their in-depth review. Our original related work section focused on immediate predecessors of our results. We agree that it makes sense to provide context by citing a broader range of related work. We will add references to work on PAC-Bayes ([Rivasplata et al., 2018], [Dziugaite & Roy, 2018], etc.), on information theoretic bounds ([Russo & Zou, 2016], [Raginsky et al., 2016], etc.) and on the Langevin algorithm ([Raginsky et al., 2017], etc.). Reviewer #2 also points out that Russo and Zou provide a correct proof for Xu–Raginsky; we now cite them for this as well. If the reviewers believe there are other articles we should cite, we would be grateful if they would could bring those to our attention.

4. Reviewer #2 points out that in Equation (31) it is not clear that the RHS is constant in $J$. Note that $S_J^c$ is a random variable that is a $(n-m)$-tuple. The statement is true since the mutual information $I(W; S_J^c)$ is a number that only depends on joint law of $(W, S_J^c)$, not on the values of $W$ or $S$ or $J$ in a particular realization.

5. Reviewer #2 requests that we compare actual generalization bounds as opposed to "key quantities". Reviewer #1 also remarks that our figure labels are not informative. We note that comparing actual generalisation bounds is made cumbersome due to the non-linearity of our bound. In particular, the form $\mathbb{E}[\sqrt{...}]$ can be upper bounded by $\sqrt{\mathbb{E}...}$ using Jensen's inequality, but this change makes a material difference in the quality of the bound. Moving as many expectation outside of the $\sqrt{\cdot}$ as possible is one of the key advantages of our work over other related work. For this reason we plotted the *sum of the gradient covariances* and *sum of gradient norms* against epoch number. We clarify our numerical results and properly label our figures in the final version. To provide evidence that our method does improve over other work and yield a non-vacuous bound, we provide Monte Carlo estimates of our bound (Eq. (22)) and that of [Mou et al., 2017] (their Eq. (69)) in the table below. The bounds are dependent on architecture, data

| | MNIST with MLP | | | MNIST with CNN | | |
|---|---|---|---|---|---|---|
| | Epoch 1 | Epoch 2 | Epoch 3 | Epoch 1 | Epoch 2 | Epoch 3 |
| Training Classification Error | $25.52 \pm 0.08\%$ | $16.17 \pm 0.04\%$ | $12.38 \pm 0.02\%$ | $21.89 \pm 0.21\%$ | $14.07 \pm 0.14\%$ | $10.78 \pm 0.10\%$ |
| Test Classification Error | $25.57 \pm 0.06\%$ | $16.29 \pm 0.04\%$ | $12.45 \pm 0.02\%$ | $22.93 \pm 0.20\%$ | $14.72 \pm 0.14\%$ | $11.24 \pm 0.09\%$ |
| Generalization Gap (Mou et al.) | $33.8 \pm 1.4\%$ | $76.0 \pm 3.0\%$ | $139.4 \pm 5.9\%$ | $46.5 \pm 2.2\%$ | $78.6 \pm 3.0\%$ | $130.6 \pm 4.6\%$ |
| Generalization Gap (Our Bound) | $10.0 \pm 1.6\%$ | $20.5 \pm 4.0\%$ | $29.0 \pm 6.7\%$ | $15.3 \pm 2.8\%$ | $25.8 \pm 4.4\%$ | $49.2 \pm 10.4\%$ |

distribution, and hyperparameters. We did *not* attempt to tune the hyperparameters to make the predictive performance or generalization bounds better. If the reviewers think it's worthwhile, we can add these results to the final version of the paper. However, we believe our current experiments are also sufficient and demonstrate the theoretical advance. Note that, in Appendix G, we already give an example where the improvement of our bound over related work can be made arbitrarily large when the data distribution is sufficiently heavy tailed, due to the order of $\mathbb{E}$ and $\sqrt{\cdot}$ yielding $\mathbb{E}|Z| \ll \sqrt{\mathbb{E}Z^2}$.

6. Reviewer #3 suggests that we state the results for LD and SGLD as formal theorems. First, we acknowledge and thank the reviewer for their in-depth review. We like this idea, thank you.

7. Reviewer #3 suggests that we should provide more intuition regarding the roles of the subsample $J$ and the auxiliary random variable $X$. The utility in keeping track of these quantities as analytical tools is one of the main contributions of our work. The subsample, $J$, is the key to getting a data-dependent bound – in order to compare the outcome of an algorithm to that of a similar algorithm run on a subset of the data we need to select such a subset, and the randomness in the selection of the subset allows us to turn this comparison into a generalization bound. In the case of SGLD, the auxiliary variable will represent the order in which the indices of our data points appear in the minibatches. In general, the auxiliary variable collects together all nuisance variables we wish to couple.

8. Reviewer #3 intuits that for SGLD the generalization bound should be of the same order as that for LD, while our work presents a seemingly slower rate. The same lower order rate for SGLD may be found in related work, such as the PAC-Bayesian bound of [Mou et al., 2018]. Proving a generalization bound of order $O(n^{-1})$ for SGLD without unrealistically strong stability assumptions is an open problem. If we consider a constant number of iterations (i.e., less than one epoch of SGLD), our theorems can yield a $O(n^{-1})$ rate, but with worse apparent dependence on the number of iterations. We did not include this result in the submitted version as we believe results for $\geq 1$ epoch are of more interest, however some stability based work essentially depends on considering only the fractional epoch regime. If the reviewer thinks that it is worthwhile, we can expand on these issues in the final version.

[Meta-Review · NeurIPS 2019]

This paper improves upon state-of-the-art information-theoretic generalization bounds for iterative algorithms using PAC-Bayes theory. When particularized to SGLD, this machinery gives a generalization bound that scales with the trace of covariance along the trajectory of the algorithm. This is a topic of current interest, and the techniques in this paper will certainly be useful for further research.